# Generate, but Verify: Reducing Hallucination in Vision-Language Models with Retrospective Resampling

**Tsung-Han Wu**[1]    **Heekyung Lee**[1,2]    **Jiaxin Ge**[1]
**Joseph E. Gonzalez**[1]    **Trevor Darrell**[1]    **David M. Chan**[1]

[1]UC Berkeley    [2]POSTECH

## Abstract

Vision-Language Models (VLMs) excel at visual understanding but often suffer from visual hallucinations, where they generate descriptions of nonexistent objects, actions, or concepts, posing significant risks in safety-critical applications. Existing hallucination mitigation methods typically follow one of two paradigms: generation adjustment, which modifies decoding behavior to align text with visual inputs, and post-hoc verification, where external models assess and correct outputs. While effective, generation adjustment methods often rely on heuristics and lack correction mechanisms, while post-hoc verification is complicated, typically requiring multiple models and tending to reject outputs rather than refine them. In this work, we introduce REVERSE, a unified framework that integrates hallucination-aware training with on-the-fly self-verification. By leveraging a new hallucination-verification dataset containing over 1.3M semi-synthetic samples, along with a novel inference-time retrospective resampling technique, our approach enables VLMs to both detect hallucinations during generation and dynamically revise those hallucinations. Our evaluations show that REVERSE achieves state-of-the-art hallucination reduction, outperforming the best existing methods by up to 12% on CHAIR-MSCOCO and 34% on HaloQuest.

🌐 **Project Page**    ○ **Code**    🤗 **Model Checkpoints/Datasets**

## 1 Introduction

Vision-Language Models (VLMs) have revolutionized visual understanding, achieving dramatic improvements in tasks like visual question-answering and image captioning, yet they still struggle with a significant limitation: visual hallucination — the tendency to describe objects that aren't actually present in the scene. Such hallucinations pose significant risks when applying VLMs to safety-critical environments, ranging from autonomous driving scenarios and decision-making to assistive technologies for the visually impaired.

To tackle these issues, researchers have generally pursued methods following one of two paradigms: *generation adjustment* or *post-hoc verification*. Generation adjustment methods focus on aligning textual outputs more closely with visual inputs by modifying the VLM's generation behavior, either in a "training-free" way (modifying the logits at decoding time) [28, 23, 24, 3, 58], or using a "training-based" strategy requiring additional supervision or custom objective functions [54, 42, 56, 41, 33, 55]. Unfortunately, these methods have no means of correcting erroneous tokens once they have been generated, and they do not leverage powerful retrospective tools such as chain-of-thought reasoning to reason about and evaluate the quality of their generation. In contrast to generation adjustment approaches, post-hoc verification methods [53, 57, 38, 49, 42] leverage large external models, such as GPT-4 [37], as verifiers to evaluate outputs *after they have been generated*. Post-hoc verifiers are accurate at predicting hallucination, but are complicated, requiring multiple models. Post-hoc verifiers often do not provide a way for the model to *correct* the hallucination but instead adopt generic refusal strategies.

In this paper, we introduce **REVERSE** (*REtrospective VERification and SElf-correction*), the first framework that unifies generation, verification, and correction within a single VLM (Figure 1), enabling on-

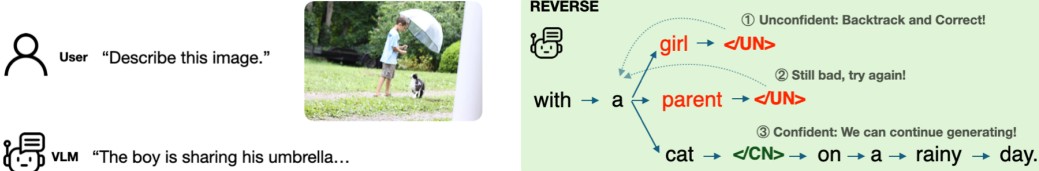

Figure 1: **REVERSE**, our proposed training and decoding paradigm for hallucination reduction, enables a single VLM to both **verify** if it has generated a hallucination and then **correct** itself iteratively. When uncertainty is detected through the generation of a (`</UN>`), the model backtracks and regenerates until a confident phrase (`</CN>`) is found.

the-fly correction during decoding without waiting for the model to complete its initial output. REVERSE consists of two key novel components. First, our method fine-tunes a VLM on a specially constructed training dataset consisting of synthetic hallucination phrases tagged with a special, explicit confidence token. Unlike prior VLMs instructed with only well-grounded data, our resulting hallucination-aware model is now able to tag likely phrase-level hallucinations during the generation process.

Second, we introduce retrospective resampling, a technique that allows the hallucination-aware VLM to serve as its own verifier. During the generation process, when the hallucination-aware VLM places sufficient probability on the special hallucination token we trigger a backtracking self-correction process. Specifically, we backtrack to a previous confident section and then apply rejection sampling and query-rewriting to correct the hallucination. As illustrated in Figure 1, the introduction of explicit confidence-token training and the backtracking-based inference algorithm enable VLMs to perform interpretable and controllable self-correction, a paradigm not explored in prior work.

We evaluate REVERSE against SOTA hallucination reduction baselines across a wide range of benchmarks designed for hallucination evaluation on LLaVA-v1.5 [34], LLaVA-MORE [17], and Qwen2.5-VL [6]. On captioning tasks, REVERSE achieves up to a 12% reduction in CHAIR scores on CHAIR-MSCOCO [40] and AMBER [47] over the best existing methods. On hallucination-sensitive open-ended tasks, it also delivers over a 10% and 34% performance improvement on MMHal [42] and HaloQuest [49], respectively.

In summary, this paper both (i) introduces REVERSE, the first hallucination reduction method unifying the generation adjustment and post-hoc verification approaches, addressing hallucination in both the training and inference stages and (ii) provides a new public training dataset and data curation pipeline for training-time hallucination mitigation consisting of 1.3M semi-synthetic samples. Together, these contributions allow REVERSE to achieve up to a 12% improvement on the CHAIR-MSCOCO benchmark, and a 34% improvement on the HaloQuest benchmark over existing SOTA methods for hallucination reduction under the same setting, especially in questions with false premise and insufficient context.

## 2  Background & Related Work

Following the success of Large Language Models (LLMs) [44, 4, 5, 32, 11, 1], Vision-Language Models (VLMs) have shown success across various multimodal tasks, such as image captioning, visual question answering, visual reasoning, and image segmentation [29, 2, 34, 35, 37, 43, 6, 52, 51, 27, 48]. Despite their impressive performance, VLMs are prone to hallucinations: generating incorrect or nonexistent visual information [30]. To address this issue, several hallucination-specific benchmarks, such as CHAIR-MSCOCO [40], AMBER [47], MMHal [42], and POPE [30], and HaloQuest [49] have been introduced. These benchmarks evaluate VLM hallucinations across both discriminative and generative tasks, with a growing trend of using VLMs for automatic visual hallucination detection.

Beyond detection, several recent methods attempt to mitigate hallucinations by adjusting a VLM's generation process. Training-free approaches primarily focus on improving decoding strategies [28, 23, 24, 3, 58]. For instance, VCD [28] employs contrastive decoding, OPERA [23] introduces a penalty term during beam search, and DoLA [16] enhances decoding by contrasting different model layers. Training-based methods, on the other hand, aim to reduce hallucinations through improved training objectives and additional data. Some approaches leverage data augmentation [10], while others refine training via reinforcement learning from human feedback (RLHF) [42, 54, 56]. Additional methods fine-tune VLMs using custom loss functions, such as EOS token-based penalties for lengthy descriptions [55], contrastive learning from paired correct-hallucination data [24], and visual instruction tuning with improved datasets [33]. However, these approaches merely adjust the generator's behavior rather than fundamentally eliminating hallucinations—once incorrect information is produced, there is no built-in mechanism for correction.

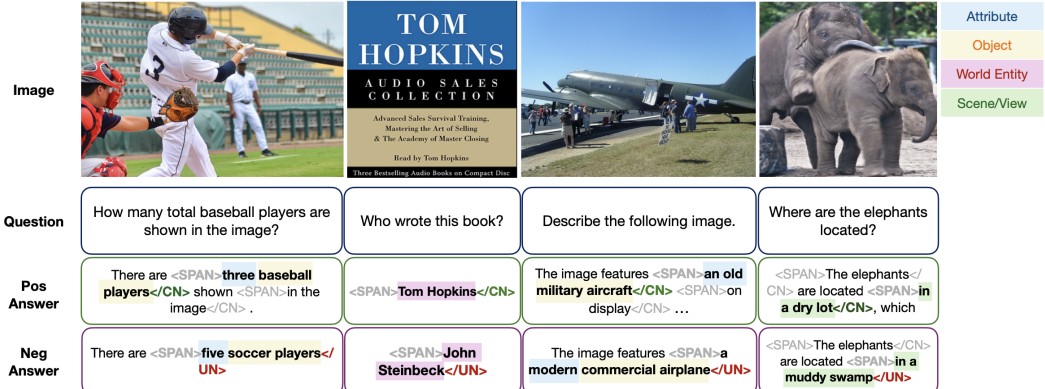

Figure 2: **Our 1.3M semi-synthetic instruction-tuning dataset for hallucination-aware VLM training.** We constructed the dataset by augmenting negative phrases from the original LLaVA-v1.5-665k [35] dataset. Our negative phrases span a diverse range, including attributes, objects, world entities, and novel scenes. Positive noun phrases are marked with `` and `</CN>`, while negative samples are enclosed with `` and `</UN>`, terminating immediately. Further details about our dataset creation and statistics can be found in subsection 3.1 and Appendix B.

The closest prior work to ours includes Woodpecker [53] and LURE [57], which use external models to verify and rewrite initial outputs from a VLM. While effective to some extent, these methods rely on complex, multi-stage pipelines with external dependencies. Moreover, they suffer from error propagation, as a single-round rewriting step is often insufficient to fully recover from low-quality initial outputs.

In contrast, our method is the first unified framework where the VLM itself serves as both the generator and verifier, enabling self-correction in a streamlined and integrated manner. Compared to prior generation-adjustment methods, our self-verification pipeline allows VLMs to retrospect and iteratively self-correct after content has been generated. Compared to existing post-hoc refinement approaches, our method eliminates the need for external models or complex multi-stage pipelines, achieving better results as the verifier can instantly and iteratively correct the generator's outputs.

# 3 REVERSE: Retrospective Verification and Self-Correction

REVERSE (*REtrospective VERification and SElf-correction*) is a hallucination reduction paradigm for Vision-Language Models (VLMs) that unifies generation adjustment and post-hoc verification methods. REVERSE allows VLMs to be hallucination-aware by explicitly modeling and monitoring the likelihood that each generated phrase is well-grounded. During training, the model is explicitly trained to classify each groundable phrase as either "confident" or "unconfident" and during inference, the model generates responses while continuously verifying the confidence of each phrase using the likelihood of the "unconfident" predictor. If a phrase is sufficiently ungrounded, the model then performs retrospective adjustment to refine the segment, enabling self-correction on the fly.

Key to the first goal of classifying each phrase as "confident" or "unconfident" is training the model to understand if a phrase is well-grounded. While VLMs and LLMs inherently provide implicit confidence scores through token probabilities, these scores are often mis-calibrated and do not consistently correlate with output correctness, making them unreliable for verification [50, 15, 18]. Furthermore, even when accurate, these probabilities offer no indication of where to backtrack for phrase re-generation and self-correction.

To overcome these limitations, we introduce three tokens to the VLM vocabulary that can be used to explicitly mark key phrases and represent the model's confidence level:

- ``: Marks the beginning of key or object phrases.
- `</CN>`: Marks the end of confident, grounded phrases.
- `</UN>`: Marks the end of unconfident, hallucinated phrases.

These tokens, when placed before/after objects or phrases in the scene can serve as ad-hoc classifiers of the confidence of the model. I.e. if a model generates a `</UN>` token after a phrase, that phrase can be considered to be ungrounded, while if it generates a `</CN>`, that phrase is likely grounded in the image. Annotating our data with such tokens, as is shown in Figure 2, will allow us to train the VLM itself to perform post-hoc verification instead of relying on an external model.

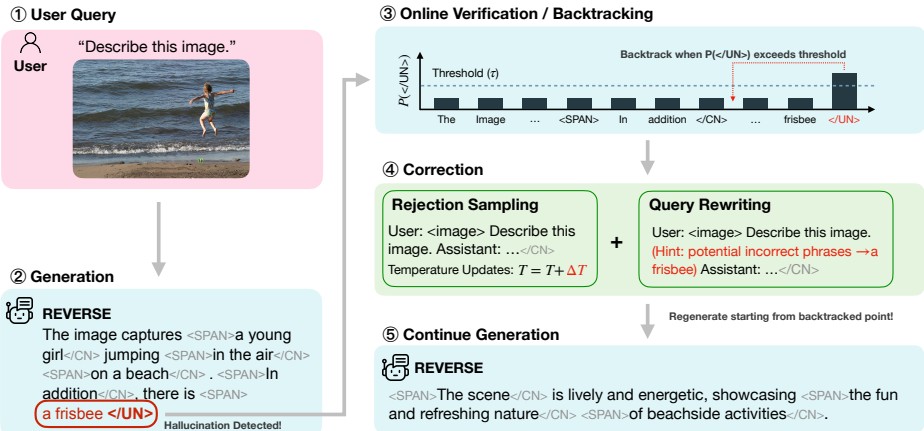

Figure 3: **Illustration of our retrospective resampling process.** During inference, we monitor the hallucination-aware VLM's generation. When the likelihood of the `</UN>` token surpasses a predefined threshold, we trigger backtracking to the most recent confident checkpoint (`</CN>`) and apply corrections using rejection sampling and query rewriting. This self-correction mechanism can be applied iteratively throughout the generation process.

## 3.1 Data Curation

Towards models that are capable of automatically tagging phrases as confident or unconfident, we constructed a 1.3M VLM instruction-tuning dataset containing a total of 6.8M question-answer pairs (turns), of which 3.8M are correct answers and 2.9M are hallucinated answers. In each response, all positive phrases are enclosed with `` and `</CN>` while the negative phrases are enclosed with `</UN>`.

The phrases in this dataset are automatically generated by annotating existing training data for VLM with these tokens. Our dataset is initially sourced from the LLaVA-v1.5-665k instruction tuning data [35], which contains only "positive" or "well-grounded" samples. To introduce "negative" or "un-grounded" samples, we designed a multi-step pipeline that generates incorrect answers leveraging rule-based algorithms and `gpt-4o-mini-0718` [37]. Specifically, we first classify the answer types, in which we can augment most of the answers with rule-based methods easily, such as binary Yes/No questions or counting questions. For the remaining general answers, mostly long answers or descriptions, we apply AI inference for high-quality and diverse data augmentation. For negative samples, constrain the sentence to immediately terminate upon reaching `</UN>`. This design not only prevents VLMs from continuing to generate ungrounded descriptions after detecting hallucinated content but also helps maintain training data quality, as any remaining context may become meaningless once the preceding information has been altered. To support retrospective query re-writing, we further inject negative keywords as hints (further discussed in subsection 3.3). Our dataset is twice the size of the LLaVA-v1.5-665k instruction tuning dataset while maintaining a similar overall composition. It preserves the same average question-answer pairs per sample and a comparable question type distribution. More details about the dataset/dataset generation pipeline are provided in Appendix B.

## 3.2 Hallucination-aware Training

To train REVERSE to recognize and respond to the new tokens, we introduce a modified cross-entropy next-token prediction loss that prevents hallucination while modeling confidence levels. Our training objectives are threefold. First, we aim to enable conventional instruction tuning to allow VLMs to perform next-token prediction to generate accurate answers. Second, we wish to reduce the likelihood of generating the hallucinated tokens that we have introduced in the new dataset. Third, we want to teach the model to generate `` at the start of key phrases, and `</CN>` or `</UN>` as explicit confidence estimators around those phrases. We achieve all of these goals by assigning a weight to each token during training; positive weights are assigned to tokens outside the `...</UN>` bounds, encouraging standard next-token prediction while zero-weights are assigned to tokens within `` and `</UN>` (i.e., masking out the targets) to avoid impacting the likelihood when training on ungrounded data (and reinforcing language priors).

Formally, let $\theta$ be our model and $D$ be the labeled VQA dataset, where each sample $S$ consists of an input sequence $X = \{x_1, x_2, ..., x_m\}$ and an output sequence $Y = \{y_1, y_2, ..., y_n\}$. Here, $X$ includes both encoded image features and question (query) tokens, while each $y_i$ in $Y$ can be either a text token corresponding to the answer or one of the three special tokens.

The model $\theta$ predicts the next token probability as $P(y_i | x_1, x_2, ..., x_m, y_1, y_2, ..., y_{i-1}; \theta)$. We then define the modified negative log-likelihood loss for a given sample as:

$$L(S) = -\sum_{y_i \in Y} \mathbb{1}_{Hall(i)} \cdot \log P(y_i | X, y_1, ..., y_{i-1}; \theta) \tag{1}$$

where $\mathbb{1}_{Hall(i)} \in \{0, 1\}$ is an indicator variable taking the value 1 for all tokens **except** those enclosed by `` and `</UN>`. For tokens within these markers, $\mathbb{1}_{Hall(i)} = 0$. Note that this masking is applied to the targets rather than the inputs. We optimize model parameters $\theta$ using the loss in Equation 1. The full training procedure is described in section 4.

## 3.3 Retrospective Resampling

During inference, the model follows standard next-token prediction but continuously monitors the likelihood of `</UN>`, triggering retrospective resampling when that likelihood exceeds a pre-defined threshold (see Figure 3). Specifically, whenever `</CN>` or `</UN>` is generated (the end of a span), we compute the probability of `</UN>`, denoted as $P(</UN>)$, across previous tokens. If $P(</UN>)$ surpasses a predefined threshold $\tau$, the model initiates a self-correction process via backtracking and retrospective resampling. Otherwise, generation proceeds normally, with ``, `</CN>` and `</UN>` tokens removed before presenting the final output.

**Backtracking Strategies**     A critical challenge in self-correction is determining both (1) where to backtrack and (2) how to regenerate content. To determine where to backtrack to, our approach follows a hierarchical fallback strategy:

1. The model first backtracks to the most recent `</CN>`, which attempts to adjust only the local information to reduce the likelihood of hallucination.

2. If the issue persists after $K$ local correction attempts, it is likely that the hallucination issue stems from earlier information in the sequence. Thus, we revert further, backtracking to the last sentence boundary (indicated by the last punctuation token).

3. If self-correction continues to fail after $N$ total attempts, the output is finalized and returned to the user, along with an indication that a hallucination was detected, but could not be corrected.

Since $P(</UN>)$ is typically low (even under hallucinations), explicitly waiting for `</UN>` to appear in the output is impractical. Instead, we set a confidence threshold $\tau$, which allows proactive identification of hallucinated phrases before they fully form. The effect of $\tau$ selection is analyzed in subsection 4.2. After the backtracking has occurred, we leverage *rejection sampling* and *query rewriting* for self-correction.

**Rejection Sampling**     Rejection sampling refines uncertain phrases by resampling multiple times at an increased temperature, seeking an alternative where $P(</UN>)$ remains below $\tau$. The process continues until reaching a confident phrase (marked by `</CN>`) or exhausting the maximum resampling attempts. In this work, we make no attempts to "ban" the generation of the same tokens during the re-sampling process. This procedure, while potentially more efficient, often leads to issues where innocuous tokens such as "a" or "the" are banned, leading to disfluencies in the final generated text. Instead, we rely on the increased temperature to lead to new candidates over several repeated generations. While rejection sampling is effective for resolving localized hallucinations, its success depends on the ability of the model to generate valid alternatives within a reasonable number of attempts. In cases where repeated resampling fails to produce an acceptable phrase, the model may need to fall back on broader correction strategies, such as query rewriting, to address deeper inconsistencies in the generated content.

**Query Rewriting**     In addition to rejection sampling, we found that "query rewriting" can provide stronger signals for VLMs to do self-correction. Query rewriting dynamically modifies the prompt to encourage better factual grounding. Specifically, the input prompt is augmented with a clarification hint:

```
<system-prompt> [<optional image>] <question> (Hint:  potential
incorrect phrases → <placeholder>)
```

This prompt signals the model to reconsider flagged segments and generate a more reliable response. In addition to rejection sampling with increased temperature, which iteratively refines outputs by resampling under varied decoding conditions, query rewriting can directly influence the model's contextual

Table 1: Performance comparison of various hallucination reduction methods across various image captioning benchmarks, which are commonly used to evaluate visual hallucinations in generative tasks for VLMs. This includes the CHAIR-MSCOCO benchmarks from [55] and the generative subset of AMBER. † and ‡ mean that we reproduced the results of these methods on CHAIR-MSCOCO and AMBER-G respectively. Otherwise, from [55] and [41].

| Base VLM | Method Type | Method | CHAIR-MSCOCO | | AMBER-G | | | |
|---|---|---|---|---|---|---|---|---|
| | | | $\text{CHAIR}_i(\downarrow)$ | $\text{CHAIR}_s(\downarrow)$ | CHAIR ($\downarrow$) | Cover ($\uparrow$) | Hall ($\downarrow$) | Cog ($\downarrow$) |
| LLaVA-v1.5 7B [35] | None | | 15.4 | 50.0 | 7.8 | 51.0 | 36.4 | 4.2 |
| | Gen-Adjust | VCD [28] | 14.9 | 48.6 | - | - | - | - |
| | | OPERA‡ [23] | 14.6 | 47.8 | 7.3 | 49.6 | 32.0 | 3.5 |
| | | DoLA† ‡ [16] | 14.1 | 51.6 | 7.6 | 51.6 | 36.0 | 4.0 |
| | | AGLA [3] | 14.1 | 43.0 | - | - | - | - |
| | | MEMVR [58] | 13.0 | 46.6 | - | - | - | - |
| | w/ Train | EOS [55] | 12.3 | 40.2 | 5.1 | 49.1 | 22.7 | 2.0 |
| | | HALVA [41] | 11.7 | 41.4 | 6.6 | **53.0** | 32.2 | 3.4 |
| | | HA-DPO [56] | 11.0 | 38.2 | 6.7 | 49.8 | 30.9 | 3.3 |
| | Post-hoc Refine | Woodpecker† [53] | 14.8 | 45.8 | 6.9 | 48.9 | 30.4 | 3.6 |
| | Combination | **REVERSE**$_{(\tau=0.003)}$ | 10.3 | 37.0 | 6.0 | 52.2 | 30.4 | 3.0 |
| | | **REVERSE**$_{(\tau=0.0003)}$ | **6.1** | **13.6** | **4.0** | 26.9 | **10.2** | **0.9** |
| LLaVA-MORE 8B [17] | | None† ‡ | 14.4 | 52.0 | 7.8 | 53.1 | 36.6 | 3.9 |
| | | DoLA† ‡ [16] | 13.8 | 51.8 | 7.9 | 53.1 | 38.4 | 4.1 |
| | | Woodpecker† ‡ [53] | 14.3 | 51.0 | 7.4 | 50.7 | 36.7 | 3.7 |
| | | **REVERSE**$_{(\tau=0.003)}$ | 12.2 | 42.4 | 6.5 | **54.8** | 35.5 | 3.9 |
| | | **REVERSE**$_{(\tau=0.0003)}$ | **8.4** | **25.2** | **5.1** | 38.9 | **20.8** | **2.1** |
| Qwen2.5-VL$^{FT}$ 3B [6] | | None† ‡ | 12.2 | 45.8 | 7.7 | **51.7** | 35.9 | 4.1 |
| | | DoLA† ‡ [16] | 14.0 | 47.6 | 9.7 | 48.1 | 31.4 | **1.9** |
| | | **REVERSE**$_{(\tau=0.01)}$ | **10.5** | **39.4** | **7.5** | 51.5 | **34.4** | 3.6 |

understanding by reformulating its input conditions. Since our training data includes hallucination-corrected phrase pairs, we randomly inject 20% of this query-rewriting prompt into the instruction-tuning process. This improves the model's ability to recognize the hint, making retrospective resampling more effective.

## 4 Experiments

**Implementation Details** We applied our method, REVERSE, on three VLM backbones: LLaVA-v1.5 (7B) [35], LLaVA-More (LLaVA with Llama-v3.1 8B) [17, 1], and Qwen2.5-VL (3B) [6]. Since LLaVA provides both its pre-trained model and instruction tuning data, we performed LoRA fine-tuning on the pre-trained model directly with our modified cross-entropy loss (see subsection 3.2) and the 1.3M-sample dataset for one epoch. In contrast, Qwen2.5-VL does not release its instruction tuning data. To enable a fair comparison, we perform full fine-tuning on the publicly available Qwen2.5-VL model using two alternatives: a 100k subset of LLaVA's instruction data and a matched subset from our dataset. Although a more direct evaluation using Qwen2.5-VL's original instruction tuning data and the augmentation method described in subsection 3.1 would be ideal, it is not feasible given the current release conditions. More details on training recipes are provided in Appendix E.

During inference, we apply retrospective resampling with different threshold values: $\tau=0.003$ for LLaVA-series models and $\tau=0.01$ for Qwen2.5-VL. These values are set per model backbone, as confidence scores across LLMs and VLMs are typically not calibrated and are rarely shared due to differences in training [25, 13]. To ensure fairness, we apply a consistent threshold per model across all evaluation datasets. Further discussion on how this controllable threshold affects model behavior is provided in subsection 4.2. For the correction mechanism, we allow up to a $N=50$ total correction attempts, with local correction attempts of $K=10$. Additionally, we implement rejection sampling with a base temperature of $T_0$, gradually increasing it with a step size of $\Delta T=0.1$, capped at a maximum temperature of $T_0+0.5$: $T=\min(T+\Delta T, T_0+0.5)$.

**Evaluation Protocol** To evaluate our approach, we compare REVERSE with various hallucination mitigation methods, including training-free or training-based generative adjustment techniques [28, 23, 16, 3, 55, 41, 56, 58], and post-hoc verification with refinement [53]. All methods are evaluated on both VLM backbones under consistent settings, where we fix the decoding temperature at 0 and use only the base prompts provided by each dataset to ensure fair comparisons. Since REVERSE does stochastic sampling at inference time, we report the mean performance over 100 bootstrapped runs for robustness. The exact numbers with 95% confidence intervals are provided in Appendix D.

Table 2: Performance on HaloQuest [49]. FP, VC, and IC stand for false premise, visually challenging, and insufficient context, three subsets in the benchmark. We ablate the effect of a lower threshold on two models and find that REVERSE improves performance on unanswerable questions without the need for specialized training.

| Method | Avg. Acc. (↑) | FP Acc. | VC Acc. | IC Acc. |
|---|---|---|---|---|
| **LLaVA-v1.5 7B** | | | | |
| None[†] | 22.6 | 17.1 | 39.5 | 10.7 |
| DoLA[†] [16] | 22.9 | 17.2 | **40.1** | 11.6 |
| HALVA[†] [41] | 23.9 | 21.1 | 37.4 | 10.7 |
| **REVERSE**$_{(\tau=0.003)}$ | 30.7 | **31.8** | 31.5 | 26.9 |
| **REVERSE**$_{(\tau=0.0003)}$ | **32.3** | 29.4 | 18.7 | **58.8** |
| **LLaVA-MORE 8B** | | | | |
| None[†] | 22.4 | 15.8 | 43.4 | 7.4 |
| DoLA[†] [16] | 22.8 | 15.5 | **45.1** | 7.4 |
| **REVERSE**$_{(\tau=0.003)}$ | 26.7 | 30.0 | 31.3 | 11.7 |
| **REVERSE**$_{(\tau=0.0003)}$ | **36.7** | **39.5** | 30.9 | **38.1** |
| **Qwen2.5-VL**[FT] **3B** | | | | |
| None[†] | 33.5 | 25.4 | **51.6** | 26.4 |
| DoLA[†] [16] | 27.4 | 16.5 | 51.1 | 19.0 |
| **REVERSE**$_{(\tau=0.01)}$ | **45.1** | **42.9** | 41.8 | **55.5** |
| GPT-4o | 63.2 | 65.2 | 55.2 | 68.7 |
| Gemini 1.5 Pro | 77.9 | 83.7 | 56.3 | 92.5 |

Table 3: Performance on MMHal-Bench [42]. Results re-implemented by us are marked with [†]. Consistent with findings on HaloQuest, applying a lower threshold ($\tau = 0.0003$) in REVERSE enables the VLM to better handle false-premise and unanswerable questions, which are common in MMHal. It achieves higher scores and lower hallucination rates, even without training on these QA pairs.

| Base VLM | Method | Score (↑) | Hall. Rate (↓) |
|---|---|---|---|
| LLaVA-v1.0 7B | LLaVA-RLHF [42] | 2.05 | 0.68 |
| LLaVA-v1.5 7B | None [35] | 2.11 | 0.54 |
| | HACL [24] | 2.13 | 0.50 |
| | HA-DPO [56] | 1.97 | 0.60 |
| | EOS [55] | 2.03 | 0.59 |
| | HALVA [41] | 2.25 | 0.54 |
| | DoLA[†] [16] | 2.33 | 0.56 |
| | Woodpecker[†] [53] | 2.19 | 0.58 |
| | **REVERSE**$_{(\tau=0.003)}$ | 2.56 | 0.47 |
| | **REVERSE**$_{(\tau=0.0003)}$ | **3.28** | **0.30** |
| LLaVA-MORE 8B | None[†] | 2.50 | 0.53 |
| | DoLA[†] [16] | 2.54 | 0.51 |
| | Woodpecker[†] [53] | 2.28 | 0.58 |
| | **REVERSE**$_{(\tau=0.003)}$ | 2.28 | 0.54 |
| | **REVERSE**$_{(\tau=0.0003)}$ | **2.93** | **0.40** |
| Qwen2.5-VL[FT] 3B | None[†] | 2.89 | 0.43 |
| | DoLA[†] [16] | 2.72 | 0.46 |
| | **REVERSE**$_{(\tau=0.01)}$ | **3.15** | **0.29** |

Our evaluation dataset covers several standard VQA tasks aimed at assessing visual hallucination, with a primary focus on image captioning and open-ended question answering. While discriminative tasks, binary (Yes/No) questions targeting object, attributes, and spatial understanding, are also common, backtracking provides limited benefit in such settings, which have been noted to offer less diagnostic insight into VLM hallucinations [7, 41]. We include results on these tasks in Appendix D for completeness.

For image captioning, we use CHAIR-MSCOCO [40, 55] and the generative subset of AMBER [47] (denoted as AMBER-G). CHAIR-MSCOCO evaluates object hallucination using the CHAIR score, which measures the degree of misalignment between objects mentioned in a model-generated caption and objects actually present in the image. It is defined as 1 minus the intersection over union (IoU) between the sets of mentioned and ground-truth objects. We report both CHAIR$_i$, which aggregates the CHAIR score across all object instances, and CHAIR$_s$, which quantifies the proportion of images where at least one hallucination occurs. For AMBER-G, we report four key metrics: CHAIR, Coverage (Cover), Hallucination (Hall), and Cognition (Cog). CHAIR is the same as CHAIR$_i$ above, and Coverage measures how well the caption mentions all objects in the image, similar to recall. The definitions of the remaining two metrics and further details on these datasets are provided in Appendix C.

For open-ended question answering, we evaluate on MMHal-Bench [42] and HaloQuest [49], which generally test VLMs on false-premise questions, questions with insufficient visual evidence, and visually complex queries. Following standard evaluation protocols, we assess MMHal-Bench responses using `gpt-4-0314` and HaloQuest with `Gemini-1.5-Pro`, as the original paper used `Gemini-1.0-Pro`, which is no longer available. These benchmarks require models to generate free-form text answers, testing their ability to comprehend and reason about visual content in an open-ended manner.

## 4.1 Experimental Results

**Image Captioning Tasks**   Table 1 presents results on image captioning tasks. With the default parameters, our method achieves the best results on LLaVA-series and Qwen2.5-VL, reducing the CHAIR$_i$ value by up to 12% on CHAIR-MSCOCO and AMBER-G compared to the best existing methods. On LLaVA-v1.5-7B, training-free methods underperform relative to fine-tuned approaches. DoLA [16] reduces hallucinations for LLaMA-based models but does not generalize well to Qwen2.5-VL. EOS [55] performs well on AMBER-G, likely because it encourages the model to produce more concise captions, leading to less informative outputs and reduced coverage. The post-hoc refinement method, Woodpecker [53], employs a multi-stage process for verification and correction; however, its one-time correction approach may suffer from error propagation, limiting its effectiveness. In contrast, REVERSE generalizes across models and supports threshold-based tuning to balance hallucination and coverage (more studies in subsection 4.2). Figure 4 presents qualitative

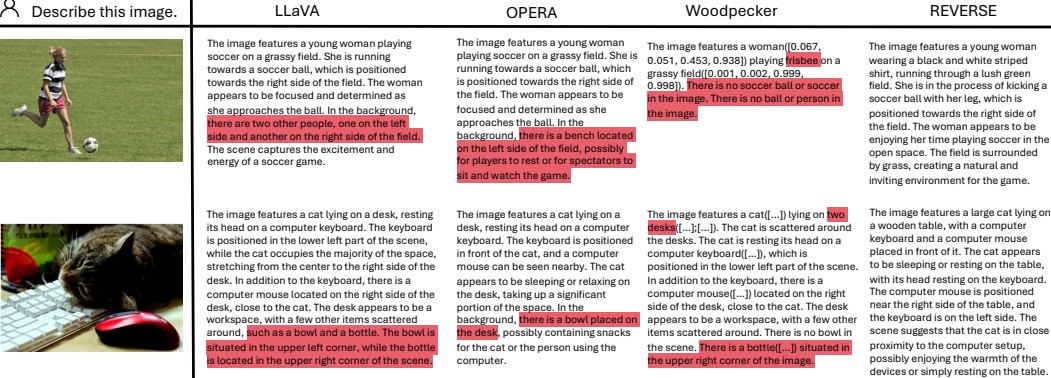

Figure 4: **Qualitative Examples of different Methods.** When generating captions for an image, LLaVA, OPERA, and Woodpecker tend to hallucinate non-existing objects. REVERSE generates correct captions of similar length. Additional qualitative results are provided in Appendix D.

results using four different methods. LLaVA-v1.5-7B, OPERA, and Woodpecker hallucinate non-existing objects, while REVERSE can generate the correct caption without reducing caption length too much.

**Open-ended Question Answering** We evaluate REVERSE on MMHal-Bench and HaloQuest, two open-ended VQA benchmarks containing lots of questions with false premises or insufficient context. These examples require the model to either refuse or correct the query. For these questions, we observed that REVERSE often produces empty responses and we interpret this behavior as the model identifying the query as unanswerable. In all such cases, we apply query rewriting with the prompt: "For this question, please point out the false premises or note what information is missing, rather than answering it directly." The complete mechanism for handling unanswerable questions is given in Appendix E.

As shown in Table 3 and Table 2, REVERSE improves accuracy by up to 10% on MMHal-Bench and 34% on HaloQuest compared with the SOTA models using default hyperparameters ($\tau=0.003$ for LLaVA series and $\tau=0.01$ for Qwen2.5-VL). Most of the gains come from better handling of false-premise and insufficient-context questions. However, performance on visually challenging questions decreases as the model adopts a more cautious approach and avoids speculative answers, even when they may be correct. Additional experiments with LLaVA show that lowering the threshold further (e.g., $\tau=0.0003$) increases conservativeness and boosts performance on ambiguous queries, without requiring task-specific fine-tuning.

## 4.2 Discussions

We conduct additional experiments on two LLaVA models to examine key aspects of our method, including ablation studies, trade-offs between expressiveness and performance, efficiency versus accuracy, and the effect of temperature. Potential limitations and broader social impacts are discussed in section 5.

**Ablation Studies** We conduct ablation studies to evaluate the contributions of different components of our method, as shown in Table 4. Comparing the first and second rows, we observe that hallucination-aware training alone already improves performance across all metrics, outperforming existing VLMs. We hypothesize that this improvement arises from the model's ability to contrast positive and negative phrases, effectively learning to distinguish between </CN> and </UN> during training—a mechanism that may be similar to DPO [39]. This finding suggests a potential research direction for future work. Interestingly, even a naive rejection sampling strategy reduces CHAIR hallucination scores by 1.2. When combined with query rewriting, the coverage improves by a further 1.2 points, indicating that rewriting helps the model explore alternative phrasing and correct itself more effectively.

**Trade-offs Between Inference Efficiency and Hallucination** To quantify the impact of REVERSE on efficiency, we analyze 1,004 samples from the AMBER-G test set. As shown in Table 5, we report the number of correction rounds, hallucination scores (CHAIR), and relative runtime, measured by the total number of generated tokens of REVERSE-v1.5-7B compared to the LLaVA-v1.5-7B baseline. To show the trade-off of correction versus efficiency, these experiments are conducted under a protocol where the model continues generation after $N$ correction attempts (rather than terminating early, as our algorithm normally would). The overall overhead can be decomposed into two components: *verification overhead* and *correction*

Table 4: Ablations: hallucination-aware training improves coverage and reduces hallucination; retrospective resampling (default $\tau=0.003$) further lowers hallucination.

| Components | CHAIR (↓) | Cover (↑) | Hall (↓) | Cog (↓) |
|---|---|---|---|---|
| LLaVA-v1.5-7B | 7.8 | 51.0 | 36.4 | 4.2 |
| + Hall-aware Training | 7.2 | **53.2** | 36.3 | 3.4 |
| + Rejection Sampling | **6.0** | 51.0 | 30.5 | **3.0** |
| + Query Rewriting | **6.0** | 52.2 | **30.4** | **3.0** |

Table 5: Efficiency Study: Increasing the number of self-correction rounds reduces hallucination (CHAIR) with the cost of total generated tokens. The reported token ratio denotes REVERSE-v1.5's token count relative to the LLaVA-v1.5-7B baseline.

| # Rounds (N) | 0 | 5 | 10 | 20 | 50 |
|---|---|---|---|---|---|
| CHAIR (↓) | 7.8 | 7.1 | 6.8 | 6.7 | 6.0 |
| #Tokens (%) | 1.00× | 1.75× | 2.05× | 2.63× | 3.05× |

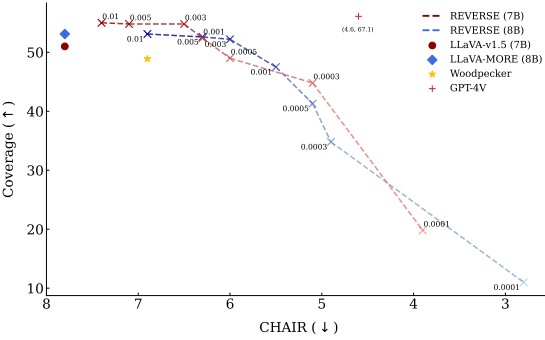

Figure 5: This plot illustrates the trade-off between CHAIR (↓) and Coverage (↑) across different threshold values. REVERSE is the first controllable VLM allowing for such tradeoffs.

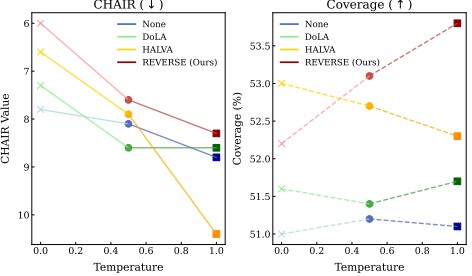

Figure 6: Increasing temperature is expected to encourage models to mention more objects and details at the cost of higher hallucination risk. REVERSE achieves the best tradeoff between expressiveness and hallucination, maintaining high coverage while surpassing all baselines on CHAIR.

*overhead*. For the verification overhead, our token-level confidence estimation is performed inline during generation and introduces negligible additional cost. In contrast, prior methods like Woodpecker [53] and LURE [57] rely on auxiliary VLMs or external object detectors, incurring substantial inference overhead.

As for the correction overhead, we further evaluate performance across different maximum correction rounds. In prior work, each correction round typically entails full re-generation, leading to approximately $(N+1)\times$ runtime for $N$ correction rounds. In REVERSE, despite setting the maximum number of correction rounds to 50, 37% of samples require no backtracking, and among the remaining cases, over half converge within a single round. This keeps the average overhead reasonable. Additional correction rounds continue to lower CHAIR scores while runtime remains within roughly $3\times$ of the baseline, since corrections are localized rather than full re-generations. Moreover, the re-generation process can be further optimized by reusing KV-cache [45, 26], avoiding recomputation of the shared prefix.

Overall, the self-correction loop in REVERSE is computationally affordable and effective, as demonstrated in Table 1, Table 2, and Table 3. Yet, we believed that developing even more efficient correction strategies remains a promising avenue for future work.

**Trade-offs Between Expressiveness and Hallucination**  As discussed in subsection 3.3, a key component of our retrospective resampling method is the predefined threshold $\tau$. When the predicted probability of `</UN>` exceeds $\tau$, the model triggers backtracking and self-correction. Figure 5 presents an analysis of the effect of $\tau$ on two VLMs. The 2D plot illustrates the performance trade-off between CHAIR (hallucination metric) and coverage across different threshold values. Based on our experiments, we selected $\tau=0.003$ as a global threshold for image captioning and other generative tasks, as it represents the peak of the performance frontier. The strong results under a single universal setting demonstrate the generalization ability of our method across diverse domains.

Moreover, the existence of threshold tuning provides an interpretable, user-adjustable control for balancing creativity and trustworthiness, which is an ability unique to our method. Compared with prior hallucination-reduction approaches, REVERSE enables dynamic adjustment of this balance between expressiveness and reliability. Notably, with a relatively high threshold ($\tau=0.01$), our method already surpasses the base VLMs (LLaVA-v1.5 and LLaVA-MORE) in both hallucination reduction and content coverage. Conversely, with a lower threshold ($\tau=0.0001$), our model can even outperform GPT-4V in hallucination control. To our knowledge, REVERSE is the first approach to expose this trade-off transparently through

a single parameter, whereas prior methods fix such thresholds implicitly without offering user control. Future work may explore how to adaptively adjust $\tau$ based on contextual factors or task difficulty.

**Impact of Temperature on Hallucination and Coverage** We also analyze how temperature settings affect hallucination rates and object coverage in generated outputs. In this experiment, we use LLaVA-v1.5-7B as the backbone and increase the number of local correction attempts ($K$) while keeping all other parameters in REVERSE unchanged. As shown in Figure 6, REVERSE is robust to increasing temperature. For tasks such as image captioning, a higher temperature is often desirable to enhance diversity in generated descriptions. However, existing methods not only suffer from increased hallucinations at higher temperatures but also exhibit a decline in object coverage. In contrast, REVERSE balances expressiveness and reliability—slightly reducing hallucination while improving object coverage as temperature increases.

## 5 Limitations and Societal Impact

**Our 1.3M Instruction Tuning Dataset** We synthesize a 1.3M-sample, hallucination-aware instruction tuning dataset. While effective as a proof-of-concept (shown in Table 1, Table 2), it has several limitations. First, as with LLaVA's dataset, ours lacks coverage of edge cases (e.g., insufficient context, false premises) and noisy labels, which is an issue recently addressed in newer datasets [14, 43]. Second, our data augmentation uses GPT-4o-mini, which may introduce bias or limited coverage. Finally, our dataset includes subsets from existing sources like MS-COCO, which contain known biases (e.g., gender, race, geography) [20, 9, 21, 46]. Future work should aim to develop more comprehensive, higher-quality datasets with reduced bias.

**REVERSE-VLMs** REVERSE is a generalizable framework applicable to models like LLaVA-v1.5, LLaVA-MORE, and Qwen2.5-VL. It significantly reduces hallucination with minimal loss in expressiveness and efficiency (shown in Table 1, Table 2). However, it does not improve performance on discriminative VQA tasks (see Table A.1) as the backtracking methods generally didn't help further reasoning. Future work can explore integrating REVERSE with existing hallucination reduction techniques to improve such tasks. As discussed in Table 5, REVERSE reduces the hallucination with the cost of more generated tokens. An efficient self-correction methods can be a future direction.

As a well studied problem, REVERSE carries risks of misuse just like other VLMs [8, 34, 35]. We adopt existing safety mechanisms from upstream models (i.e., LLaVA-v1.5 [35], LLaVA-MORE [17], and Qwen2.5-VL [6]), and provide a fully open-source release with training details.

## 6 Conclusion

In this paper, we introduced REVERSE, a framework that reduces hallucinations in Vision-Language Models by combining hallucination-aware training with retrospective resampling. REVERSE achieves up to 12% improvement on image CHAIR-MSCOCO and 34% on HaloQuest over existing SOTA methods. While preliminary, REVERSE highlights the potential of self-correction in multimodal models. Future work may explore integrating structured verification and causal reasoning to further reduce hallucinations. As multimodal AI progresses, we see self-verification and retrospective techniques as a promising direction for building more trustworthy systems.

**Acknowledgments**

We thank Shalini Ghosh and Konpat Preechakul for their invaluable feedback during the discussions. Authors, as part of their affiliation with UC Berkeley, were supported in part by the National Science Foundation, US Department of Defense, and/or the Berkeley Artificial Intelligence Research (BAIR) industrial alliance program, as well as gifts from Amazon. Sky Computing Lab is supported by gifts from Accenture, AMD, Anyscale, Cisco, Google, IBM, Intel, Intesa Sanpaolo, Lambda, Microsoft, NVIDIA, Samsung SDS, SAP, and VMware. This research was also developed with funding from the Defense Advanced Research Projects Agency (DARPA) under Contract No(s). FA8650-23-C-7316 and HR0011-25-3-0133. The views, opinions and/or findings expressed are those of the author and should not be interpreted as representing the official views or policies of any sponsor, the Department of Defense, or the U.S. Government.

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

## Appendix

The appendix consists of the following further discussion:

- Appendix A provides links to the released code, model checkpoints, and dataset.
- Appendix B describes the dataset that we constructed containing (</UN>) and (</CN>) tokens.
- Appendix C describes the evaluation datasets and metrics.
- Appendix D provides more qualitative and quantitative results.
- Appendix E details the training and evaluation implementation for REVERSE.

## A    Code and Model Release

The project website is available at: https://reverse-vlm.github.io. The code for REVERSE is released under the MIT license at https://github.com/tsunghan-wu/reverse_vlm. It builds upon the Apache 2.0-licensed codebases of LLaVA [34] and LLaVA-MORE [17], as well as the Qwen license associated with Qwen2.5-VL [6].

We also release model checkpoints of REVERSE (based on LLaVA-v1.5, LLaVA-MORE, and Qwen2.5-VL), along with a 1.3M-sample semi-synthetic dataset, at Hugging Face. Both the checkpoints and dataset are released under the MIT license. The dataset contains elements adapted from LLaVA [34], which is licensed under the Creative Commons Attribution 4.0 International License. Use of the dataset complies with OpenAI's usage policy.

## B    Dataset Details

The philosophy of our data generation is to be *automatic*, *scalable*, and *high-quality*. The overall pipeline is illustrated in Figure A.1. The dataset construction begins with the automatic annotation of all noun phrases (with multilingual support) and a set of predefined task-relevant keywords such as "Yes" or "No." To avoid uninformative annotations (e.g., "in the image"), we maintain a skip list (Figure A.3). Noun phrases and their prefix prepositions are extracted automatically using Part-of-Speech (POS) tagging tools [22]. This stage produces positive examples that accurately describe the visual content (objects). To teach the model to produce unconfident tokens (</UN>) after incorrect, hallucinated objects, we augment each positive example with one corresponding negative sample, generated through a combination of rule-based algorithms and LLM prompting with a strong emphasis on diversity and quality, as follows.

**Type-aware rules.**    We first categorize QA pairs into different types. For simple cases (e.g., yes/no or numerical questions), we apply deterministic transformations such as flipping answers or sampling plausible but incorrect numbers.

**LLM-generated negatives.**    For more complex cases (e.g., open-ended descriptive answers), we use GPT-4o to generate diverse and semantically plausible negative responses, following the structured prompts described in the following pages.

**Quality control.**    To ensure high data quality, we filter out trivial negatives (e.g., those differing only by pronoun swaps) and re-distribute examples to balance the positive/negative ordering, preventing positional bias and ensuring that each negative sample is both meaningful and informative.

The final image distribution matches that of the original LLaVA-v1.5-665k dataset, as shown in Figure A.2. Compared to existing datasets such as FAVA [36], the data used for REVERSE is fundamentally different in both scale and diversity: REVERSE contains 1.3M samples (versus 35K in FAVA) and covers a much wider range of hallucination types across attributes, objects, world entities, and scenes, extending beyond the graph-based replacement frameworks adopted in prior work.

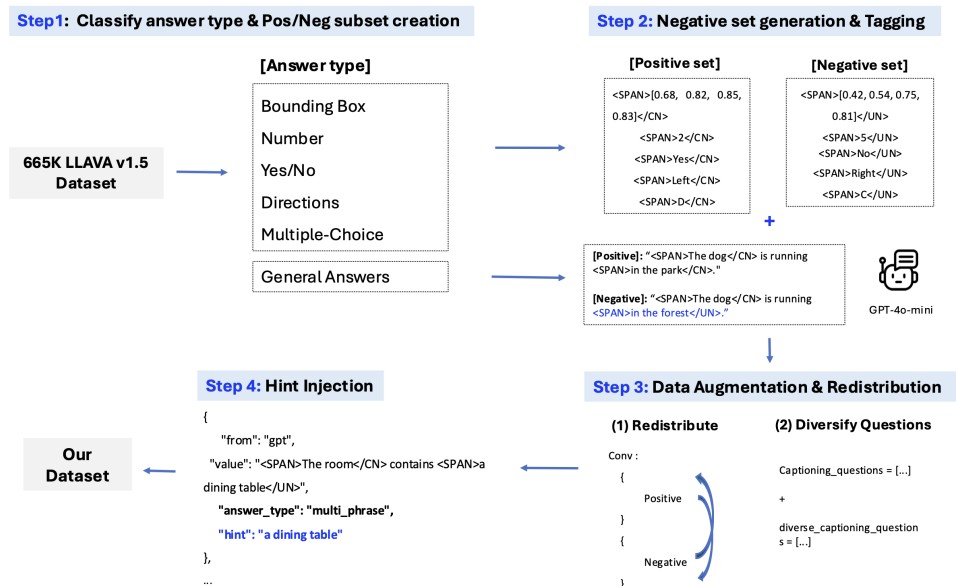

Figure A.1: The data generation pipeline for our 1.3M instruction-tuning data.

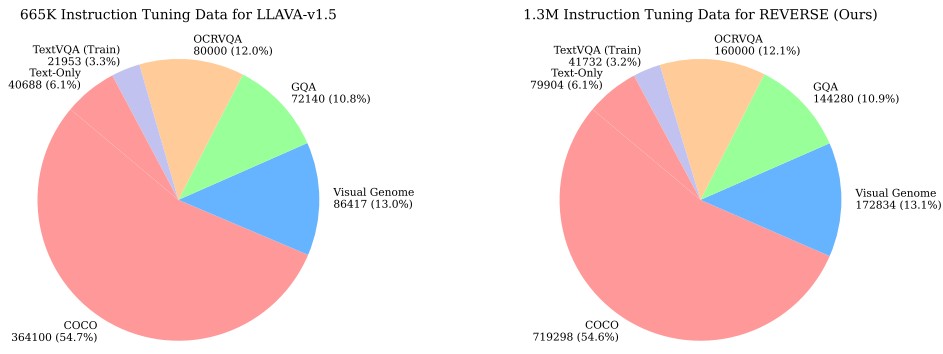

Figure A.2: Comparison between LLaVA-v1.5-665k and our 1.3M instruction tuning dataset

what, where, which, who, whom, whose, why, how,
What, Where, Which, Who, Whom, Whose, Why, How,
that, this, these, those, That, This, These, Those,
he, she, it, we, you, they, me, him, her, us, them, I,
He, She, It, We, You, They, Me, Him, Her, Us, Them, I,
my, your, his, her, its, our, their, mine, yours, ours, theirs,
My, Your, His, Her, Its, Our, Their, Mine, Yours, Ours, Theirs,
a, an, the, A, An, The,
in the image, the image, The image, In the image,
in the picture, the picture, The picture, In the picture.

Figure A.3: Words that we skip during data processing to ensure high-quality hallucination generation

## C Evaluation Datasets & Metrics

To evaluate how REVERSE reduces visual hallucination through backtracking, we use two image captioning benchmarks and two open-ended VQA datasets. While not the main focus of this paper, we also report results on standard discriminative VQA benchmarks commonly used for hallucination evaluation. Below, we describe each of these benchmark datasets in detail.

**CHAIR-MSCOCO:** MS COCO (Microsoft Common Objects in Context) [31] is a large-scale dataset designed for object detection, segmentation, and captioning tasks in computer vision. It contains over 330,000 images, with more than 200,000 labeled images spanning 80 object categories. The dataset includes detailed instance annotations, allowing for precise object localization and segmentation. Additionally, MS COCO provides five human-generated captions per image, making it a popular benchmark for image captioning and vision-language models (VLMs).

The CHAIR-MSCOCO benchmark was first introduced by Rohrbach et al. [40], which uses the full MSCOCO validation set to evaluate hallucination in vision-language models using the CHAIR score. In this work, we follow the evaluation protocol of Yue et al. [55] and assess a subset of 500 captions for efficient benchmarking.

**AMBER:** AMBER (An LLM-free Multi-dimensional Benchmark for MLLMs Hallucination Evaluation) [47] is a comprehensive evaluation framework designed to assess hallucination phenomena in Multi-modal Large Language Models (MLLMs). Unlike previous benchmarks that often rely on human or advanced LLM evaluations, AMBER offers an automated, low-cost approach to evaluate both generative and discriminative tasks. It addresses three key dimensions of hallucination: existence, attribute, and relation.

In this work, we refer to the generative subset as AMBER-G and the discriminative subset as AMBER-D. The AMBER-D subset comprises binary Yes/No questions and is evaluated using the F1 score, consistent with the POPE benchmark. Evaluation on AMBER-G is more complex and involves multiple metrics, as described below.

*CHAIR* [40] measures the percentage of hallucinated objects in a scene, and is defined as:

$$CHAIR(R) = 1 - \frac{len(R'_{obj} \cap A_{obj})}{len(R'_{obj})}. \tag{A.2}$$

where $A_{obj} = \{obj_1^A, obj_2^A, ..., obj_n^A\}$ is an annotated list of objects, and $R_{obj} = \{obj_1^R, obj_2^R, ..., obj_n^R\}$ are nouns extracted from the captions using NLTK.

*Cover* measures the object coverage of responses, namely, the proportion of objects mentioned in the response $R'_{obj}$ relative to the objects identified in the $A_{obj}$, and is defined by:

$$Cover(R) = \frac{len(R'_{obj} \cap A_{obj})}{len(A_{obj})}. \tag{A.3}$$

*Hal* is a more general metric, measuring the portion of responses containing hallucinations (similar to $CHAIR_s$). It is defined as:

$$Hal(R) = \begin{cases} 1 & \text{if } CHAIR(R) \neq 0, \\ 0 & \text{otherwise.} \end{cases} \tag{A.4}$$

*Cog* measures how similar the hallucinations that a model generates are to human hallucinations. It is defined as:

$$Cog(R) = \frac{len(R'_{obj} \cap H_{obj})}{len(R'_{obj})}. \tag{A.5}$$

for a set of target hallucinatory objects $H_{obj} = \{obj_1^H, obj_2^H, ..., obj_n^H\}$.

**HaloQuest:** HaloQuest [49] is a VQA dataset designed to evaluate and mitigate hallucination in VLMs. The evaluation set consists of over 600 examples, featuring both real images from the OpenImages dataset and synthetic images generated using tools like Midjourney. The dataset focuses on three categories of questions: those with false premises, those lacking sufficient context, and visually challenging ones, aiming to trigger common hallucination scenarios in VLMs. Performance on Haloquest is measured using "accuracy" as evaluated by Gemini-1.0 Pro; however since this model is no longer available, we leverage Gemini 1.5-Pro in this paper.

**MMHal-Bench:** MMHal-Bench [42] is an evaluation benchmark specifically designed to assess hallucination phenomena in Large Multimodal Models (LMMs). It comprises 96 challenging image-question pairs sourced from the OpenImages dataset, each accompanied by corresponding ground-truth answers and detailed image content annotations. The benchmark focuses on penalizing hallucinations by evaluating model responses against these ground truths. Performance on MMHal-Bench is measured using "score" (ranging from 0-6) and "hallucination rate" (ranging from 0-1) as evaluated by a GPT model.

Table A.1: Performance comparison across multiple discriminative visual hallucination benchmarks including the discriminative subset of AMBER (F1), POPE (F1), and the hallucination subset of MME (Score).

| Base VLM | Method | AMBER-D | POPE | MME-Hall |
|---|---|---|---|---|
| LLaVA-v1.5 7B [35] | None | 74.7 | 85.9 | 648.3 |
| | VCD [28] | - | 84.5 | 604.7 |
| | EOS [55] | 75.6 | 86.0 | 606.7 |
| | OPERA$^{\dagger}$ [23] | 74.8 | 85.5 | 592.3 |
| | DoLA$^{\dagger\,\ddagger\,\ddagger}$ [16] | 74.5 | 85.7 | 656.7 |
| | HA-DPO [56] | 78.1 | **86.9** | 618.3 |
| | MEMVR [58] | - | 85.9 | 648.3 |
| | AGLA [3] | - | 86.0 | 640.0 |
| | HALVA [41] | **83.4** | 84.8 | **665.0** |
| | Woodpecker [53] | 67.0 | - | 366.7 |
| | **REVERSE**$_{(\tau=0.5)}$ | 74.2 | 85.9 | 601.6 |
| LLaVA-MORE 8B [17] | None$^{\dagger\,\ddagger}$ | 71.6 | 85.1 | 678.3 |
| | DoLA$^{\dagger\,\ddagger\,\ddagger}$ [16] | **72.0** | **85.2** | **683.3** |
| | **REVERSE**$_{(\tau=0.5)}$ | 69.3 | 84.4 | 657.6 |
| Qwen2.5-VL$^{FT}$ 3B [6] | None$^{\dagger\,\ddagger\,\ddagger}$ | 87.7 | **87.1** | 550.4 |
| | DoLA$^{\dagger\,\ddagger\,\ddagger}$ [16] | **89.0** | 78.8 | 555.6 |
| | **REVERSE**$_{(\tau=0.5)}$ | 85.7 | 86.5 | **589.5** |

Table A.2: Additional breakdown analysis on the POPE benchmark. REVERSE achieves comparable performance to LLaVA-v1.5-7B across all subsets. Scores are reported as accuracy (95% CI).

| Model | Popular | Adversarial | Random | All |
|---|---|---|---|---|
| LLaVA-v1.5-7B | 86.1 (84.8–87.4) | 84.2 (82.8–85.7) | 87.2 (85.8–88.5) | 85.9 (83.2–88.2) |
| REVERSE | 86.3 (85.0–87.7) | 83.9 (82.4–85.3) | 87.5 (86.3–88.8) | 85.9 (82.8–88.5) |

**POPE:** The Polling-based Object Probing Evaluation (POPE) [30] is a benchmark designed to assess object hallucination in vision-language models (VLMs). Unlike traditional instruction-based evaluations, POPE employs a polling-based query method, prompting LVLMs with simple yes-or-no questions about the presence of specific objects in images. This approach converts the evaluation into a binary classification task, allowing for more stable and flexible assessment of object hallucination. Performance on POPE is measured in F1 score following LLaVA's standard [34].

**MME-Hall:** MME-Hall [19] is a specialized subset of the Multimodal Large Language Model Evaluation (MME) benchmark, focusing specifically on assessing object-related hallucinations in multimodal large language models (MLLMs). It evaluates models across four key dimensions: object existence, counting, positional accuracy, and color recognition. Most questions in this subset require binary Yes/No answers or brief responses in short phrases.

# D  Additional Results

**Discriminative Tasks:** We report results on discriminative hallucination benchmarks including AMBER-D, POPE, and MME-Hall in Table A.1 and Table A.2. These benchmarks consist of binary classification tasks such as Yes or No questions, where the impact of retrospective resampling is naturally limited. Since the answer space is minimal, often restricted to a single token, rethinking the output using a clarification hint like "(potential incorrect phrases → Yes/No)" described in subsection 3.3 generally does not provide meaningful improvement.

In this setting, we set the hallucination detection threshold to $\tau = 0.5$ for computational efficiency. Increasing the threshold beyond 0.5 does not significantly affect the results. Across all datasets, REVERSE performs comparably to existing baselines but does not show substantial gains. This is likely due to the binary nature of these tasks, which offer limited opportunity for backtracking or resampling to influence the outcome.

By contrast, for open-ended and captioning tasks, we adopt lower thresholds (0.003 for LLaVA-based models and 0.01 for Qwen2.5-VL) to account for the much larger answer space. For instance, in a sentence such as "There is a ______," the blank can be filled with many possible words, and both non-hallucinatory tokens generally have relatively low probabilities. In such cases, setting an appropriately low threshold is critical for effective hallucination detection. As discussed in section 4, we use a fixed threshold per model across datasets to ensure a fair comparison. We further reflect on its effectiveness in the discussion subsection. We leave further exploration of threshold calibration and reasoning strategies for binary tasks to future work.

Table A.3: Performance on standard VQA benchmarks. REVERSE maintains comparable accuracy to the base model across general-purpose VQA datasets.

| Method | HallusionBench | GQA | MM-Vet |
|---|---|---|---|
| LLaVA-v1.5-7B | 46.94 | 62.00 | 31.10 |
| REVERSE$_{(\tau=0.5)}$ | 45.44 | 62.73 | 28.40 |

Table A.4: Performance on captioning metrics. REVERSE significantly reduces hallucination rates while maintaining competitive caption quality.

| Method | CHAIR$_i(\downarrow)$ | CHAIR$_s(\downarrow)$ | CLAIR$(\uparrow)$ |
|---|---|---|---|
| LLaVA-v1.5-7B | 15.4 | 50.0 | 0.7384 |
| REVERSE$_{(\tau=0.003)}$ | 10.3 | 37.0 | 0.7264 |

Table A.5: Bootstrapped results on CHAIR-MSCOCO and AMBER-G. We report mean scores with 95% confidence intervals as subscripts.

| Base VLM | Method | CHAIR-MSCOCO | | AMBER-G | | | |
|---|---|---|---|---|---|---|---|
| | | CHAIR$_i(\downarrow)$ | CHAIR$_s(\downarrow)$ | CHAIR $(\downarrow)$ | Cover $(\uparrow)$ | Hall $(\downarrow)$ | Cog $(\downarrow)$ |
| LLaVA-v1.5 7B [35] | Base VLM | 15.4 | 50.0 | 7.8 | 51.0 | 36.4 | 4.2 |
| | REVERSE$_{(\tau=0.003)}$ | 10.3$_{(8.94-11.68)}$ | 37.0$_{(32.90-41.50)}$ | 6.0$_{(5.4-6.5)}$ | 52.2$_{(51.1-53.5)}$ | 30.4$_{(27.8-32.9)}$ | 3.0$_{(2.5-3.4)}$ |
| | REVERSE$_{(\tau=0.0003)}$ | 6.1$_{(4.53-7.60)}$ | 13.6$_{(10.80-16.71)}$ | 4.0$_{(2.3-5.9)}$ | 26.9$_{(24.1-30.6)}$ | 10.2$_{(6.6-14.2)}$ | 0.9$_{(0.4-1.5)}$ |
| LLaVA-MORE 8B [17] | Base VLM | 14.4 | 52.0 | 7.8 | 53.1 | 36.6 | 3.9 |
| | REVERSE$_{(\tau=0.003)}$ | 12.2$_{(10.55-13.81)}$ | 42.4$_{(38.09-46.02)}$ | 6.5$_{(5.9-7.1)}$ | 54.8$_{(53.6-56.2)}$ | 35.5$_{(32.4-38.9)}$ | 3.9$_{(3.3-4.4)}$ |
| | REVERSE$_{(\tau=0.0003)}$ | 8.4$_{(7.16-10.06)}$ | 25.2$_{(21.39-28.60)}$ | 5.1$_{(4.5-5.6)}$ | 38.9$_{(37.3-40.4)}$ | 20.8$_{(18.6-23.2)}$ | 2.1$_{(1.7-2.5)}$ |
| Qwen2.5-VL$^{FT}$ [6] | Base VLM | 12.2 | 45.8 | 7.7 | 51.7 | 35.9 | 4.1 |
| | REVERSE$_{(\tau=0.01)}$ | 10.5$_{(9.22-11.90)}$ | 39.4$_{(35.40-43.91)}$ | 7.5$_{(6.6-8.1)}$ | 51.5$_{(50.2-52.8)}$ | 34.4$_{(31.5-37.2)}$ | 3.6$_{(3.1-4.2)}$ |

**Bootstrapped Evaluation Results:** As described in section 4, we apply 100-round bootstrapping to account for variability introduced by sampling during inference, particularly for REVERSE and smaller datasets such as MM-Hal, which contains only 96 samples. Results are reported in Table A.5, Table A.6, and Table A.7.

Across all captioning and open-ended VQA tasks, REVERSE consistently demonstrates strong robustness and significantly outperforms the base VLM, with margins exceeding the 95% confidence interval. For discriminative tasks, the performance of REVERSE mostly remains within the 95% confidence interval of the base model, indicating comparable performance.

**Qualitative Examples:** Figure A.4 presents additional qualitative comparisons between REVERSE and prior methods. Words highlighted in red indicate hallucinations. Similar to Figure 4, REVERSE significantly reduces hallucinations. In particular, the fourth row clearly illustrates that when the model is uncertain, it avoids adding speculative or unsupported content.

**Results on Common VQA Tasks:** To verify that REVERSE preserves performance on standard VQA benchmarks, we further evaluate it on HallusionBench, GQA, and MM-Vet (Table A.3). For captioning tasks, we also aim to ensure that REVERSE maintains caption quality. AMBER-G already reports a "coverage" metric that measures how many relevant objects are mentioned, alongside the hallucination metric CHAIR. For evaluation on MSCOCO, we additionally report the CLAIR [12] metric, which employs an LLM-as-a-judge and correlates more strongly with human judgments than traditional automatic metrics such as CIDEr and SPICE. CLAIR scores range from 0 to 1, with higher values indicating better caption quality. The results are summarized in Table A.4.

On general VQA benchmarks, REVERSE achieves performance comparable to the LLaVA-v1.5-7B baseline, showing that the gains on hallucination-focused benchmarks (e.g., MM-Hal or Haloquest) do not come at the expense of general VLM capabilities. Although HallusionBench is sometimes considered a visual hallucination benchmark, it primarily evaluates robustness to visually misleading inputs rather than language-prior or bias-driven object hallucination that our work explicitly targets. Overall, REVERSE achieves a ~30% reduction in object hallucination while maintaining comparable caption quality to the baseline.

# E Implementation Details

## E.1 Training Details

For both LLaVA-v1.5-7B and LLaVA-MORE, we initialize from pretrained language models (Vicuna-1.5-7B and Llama-3.1-8B-Instruct, respectively), along with their corresponding visual projectors and CLIP-ViT-L/14-336 vision encoders. Following the standard LLaVA setup, we perform instruction fine-tuning using our 1.3M multi-image dataset for one epoch with LoRA (rank = 128, $\alpha = 256$). We

Table A.6: Bootstrapped results on MMHal and HaloQuest. 95% confidence intervals are shown as subscripts.

| Backbone | Method | MMHal | | HaloQuest | | | |
|---|---|---|---|---|---|---|---|
| | | Score (↑) | Hall. Rate (↓) | Avg Acc. (↑) | FP Acc. (↑) | VC Acc. (↑) | IC Acc. (↑) |
| LLaVA-v1.5 7B | Base VLM | 2.11 | 0.54 | 22.6 | 17.1 | 39.5 | 10.7 |
| | REVERSE$_{(\tau=0.003)}$ | 2.56$_{(2.22–3.01)}$ | 0.47$_{(0.35–0.56)}$ | 30.7$_{(27.4–35.2)}$ | 31.8$_{(27.5–35.7)}$ | 31.5$_{(25.2–38.1)}$ | 26.9$_{(20.8–34.7)}$ |
| | REVERSE$_{(\tau=0.0003)}$ | 3.28$_{(2.86–3.72)}$ | 0.30$_{(0.20–0.40)}$ | 32.3$_{(29.4–36.5)}$ | 29.4$_{(25.1–35.7)}$ | 18.7$_{(13.5–24.6)}$ | 58.8$_{(50.0–67.6)}$ |
| LLaVA-MORE 8B | Base VLM | 2.50 | 0.53 | 22.4 | 15.8 | 43.4 | 7.4 |
| | REVERSE$_{(\tau=0.003)}$ | 2.28$_{(1.96–2.68)}$ | 0.54$_{(0.44–0.62)}$ | 26.7$_{(23.4–30.0)}$ | 30.0$_{(26.4–35.2)}$ | 31.3$_{(25.3–38.2)}$ | 11.7$_{(6.1–16.0)}$ |
| | REVERSE$_{(\tau=0.0003)}$ | 2.93$_{(2.53–3.22)}$ | 0.40$_{(0.31–0.51)}$ | 36.7$_{(33.9–39.8)}$ | 39.5$_{(34.8–45.0)}$ | 30.9$_{(26.6–37.2)}$ | 38.1$_{(31.6–46.9)}$ |
| Qwen2.5-VL$^{FT}$ 3B | Base VLM | 2.89 | 0.43 | 33.5 | 25.4 | 51.6 | 26.4 |
| | REVERSE$_{(\tau=0.01)}$ | 3.15$_{(2.80–3.45)}$ | 0.29$_{(0.21–0.40)}$ | 45.1$_{(40.9–48.9)}$ | 42.9$_{(36.5–48.6)}$ | 41.8$_{(34.8–49.1)}$ | 55.5$_{(44.6–65.0)}$ |

Table A.7: Bootstrapped results on discriminative hallucination benchmarks. We report means with 95% confidence intervals.

| Backbone | Method | AMBER-D | POPE | MME-Hall |
|---|---|---|---|---|
| LLaVA-v1.5 7B | Base VLM | 74.7 | 85.9 | 648.3 |
| | REVERSE | 74.2$_{(73.4–75.2)}$ | 85.9$_{(82.8–88.5)}$ | 601.60$_{(555.00–642.54)}$ |
| LLaVA-MORE 8B | Base VLM | 71.6 | 85.1 | 678.3 |
| | REVERSE | 69.3$_{(68.5–70.3)}$ | 84.4$_{(81.8–86.9)}$ | 657.63$_{(615.71–696.83)}$ |
| Qwen2.5-VL$^{FT}$ 3B | Base VLM | 85.0 | 87.1 | 550.4 |
| | REVERSE | 85.7$_{(85.1–86.1)}$ | 86.5$_{(84.3–88.8)}$ | 589.48$_{(544.92–635.00)}$ |

adopt the modified cross-entropy loss defined in subsection 3.2 and train using the AdamW optimizer. The learning rate is set to 2e-5 for the visual projector, and (2e-4, 1e-4) for the LoRA parameters of LLaVA-v1.5-7B and LLaVA-MORE, respectively. The CLIP backbone is kept frozen. We use a global batch size of 128 with no gradient accumulation. Training takes 24 hours for LLaVA-v1.5-7B and 36 hours for LLaVA-MORE on 8× A100 80GB GPUs using DeepSpeed ZeRO-2.

For Qwen2.5-VL, since the instruction tuning dataset is unavailable, we finetune the released model directly. To enable apples-to-apples comparison, we apply our REVERSE finetuning on the same 100k subset used for LLaVA-FT. Unlike the LLaVA variants, we finetune the full 3B Qwen2.5-VL model (without LoRA) using the same modified cross-entropy loss. We use the AdamW optimizer with a learning rate of 5e-5, freeze the CLIP encoder, and set the batch size to 128 with no gradient accumulation. Training takes 3 hours on 4× A100 80GB GPUs using DeepSpeed ZeRO-3.

### E.2 Full Decoding Algorithm

The full decoding algorithm used in REVERSE is shown in Algorithm 1. For captioning and discriminative tasks, we directly apply this algorithm.

For open-ended tasks such as MMHal-Bench and HaloQuest, as described in section 4, we adopt a two-stage decoding process. In the first round, REVERSE runs the standard retrospective resampling algorithm. Since many of these queries contain false premises or lack sufficient information, the model is expected to abstain from answering and return a blank response. In such cases, we initiate a second round of inference with the query modified as:

```
Q := Q + "For this question, please point out the false premises
or note what information is missing, rather than answering it
directly."
```

This prompting strategy requires no additional training and enables the model to handle underspecified or invalid queries more effectively.

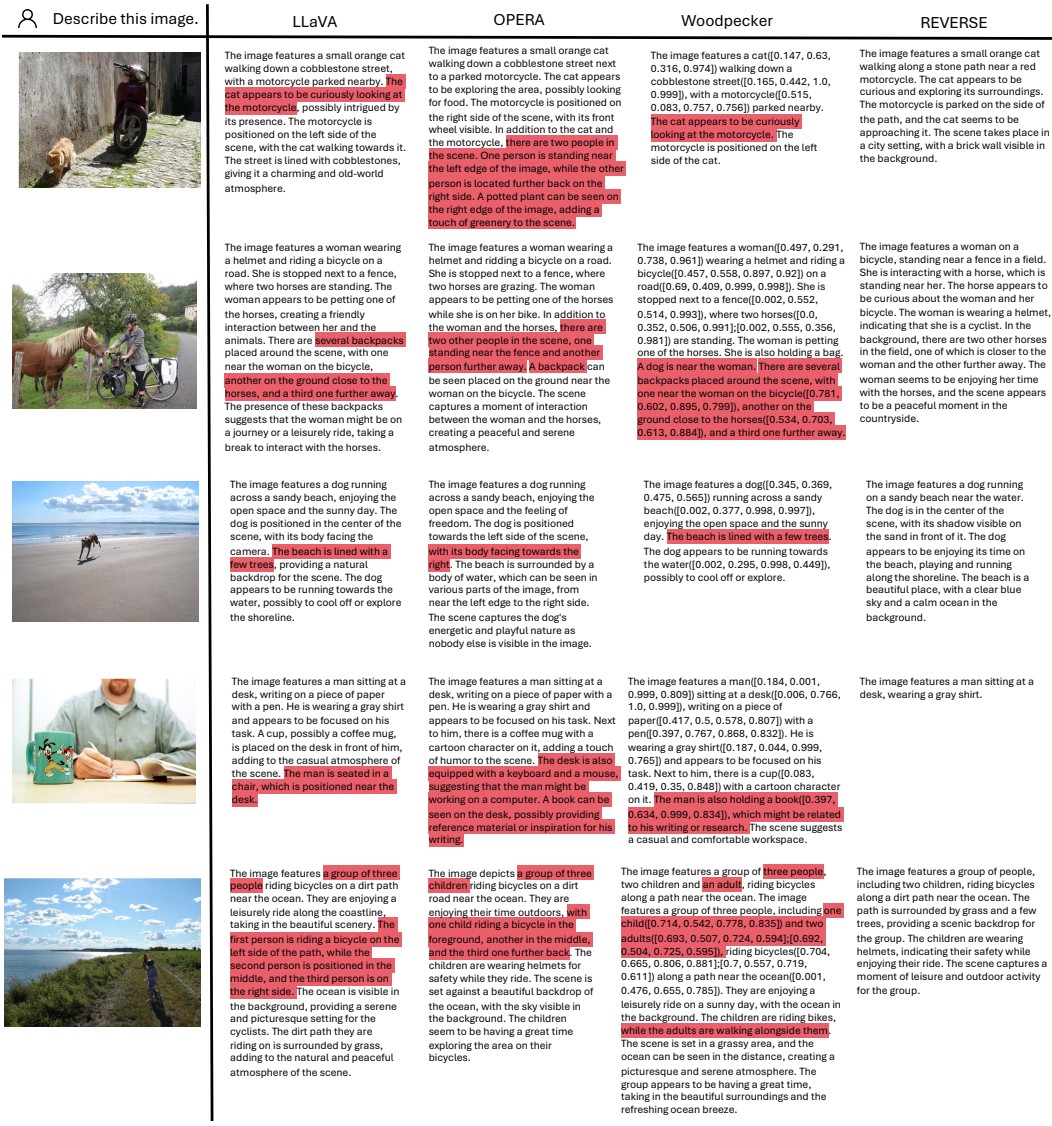

Figure A.4: Additional Qualitative Results.

**Algorithm 1** On-the-Fly Retrospective Resampling During Generation

---

**Require:** Input prompt $\{I,Q\}$ (image and question), maximum total corrections $N$, local correction
   threshold $K$, base temperature $T_0$, temperature step $\Delta T = 0.05$
1: **Initialize:** Attempt count $n \leftarrow 0$, local failures $k \leftarrow 0$, temperature $T \leftarrow T_0$, placeholder list $P \leftarrow \emptyset$
2: Initialize empty sequence $S$
3: **while** generation is not finished **do**
4:    Generate next token: $w_t = \text{VLM}_\theta(\{I,Q\} + S, T)$
5:    Append $w_t$ to $S$
6:    **if** $P(w_t) \geq \tau$ **then**                          ▷ Hallucination detected
7:       Identify most recent `</CN>` as local checkpoint $C_{\text{local}}$
8:       Append hallucinated phrase $w_t$ to placeholder list $P$
9:       **while** $n < N$ **do**                          ▷ Inside the correction loop
10:          Backtrack to $C_{\text{local}}$ and apply **Rejection Sampling and Query Rewriting**
11:          Update temperature: $T \leftarrow \min(T + \Delta T, T_0 + 0.5)$
12:          Modify prompt with clarification hint:

$$Q = Q + (\text{Hint: potential incorrect phrases} \rightarrow P)$$

13:          Generate resampled phrase: $h_t = \text{VLM}_\theta(\{I,Q\} + S, T)$
14:          **if** $P(h_t) < \tau$ for all tokens in $h_t$ **then**          ▷ Verify all token probabilities
15:             **Accept** $h_t$ and continue generation
16:             Append $h_t$ to $S$
17:             **Reset temperature:** $T \leftarrow T_0$, **Reset failure count:** $k \leftarrow 0$
18:             **Break** out of correction loop
19:          **else**
20:             $k \leftarrow k + 1$                          ▷ Track consecutive failures
21:          **end if**
22:          $n \leftarrow n + 1$
23:          **if** $k \geq K$ **then**                    ▷ Escalate backtracking if local corrections fail
24:             Identify last sentence boundary (last punctuation token) as $C_{\text{global}}$
25:             Backtrack to $C_{\text{global}}$ and reset $k \leftarrow 0$
26:          **end if**
27:       **end while**
28:    **end if**
29: **end while**
30: **Return** generated sequence $S$

---

