# OpenReview forum: "Generate, but Verify: Reducing Hallucination in Vision-Language Models with Retrospective Resampling"
_NeurIPS.cc/2025/Conference — NeurIPS 2025 poster_

### Official Review · Reviewer_B5Fh · 2025-06-04

**Clarity:** 2
**Significance:** 2
**Originality:** 2
**Rating:** 4
**Confidence:** 4

**Summary:**

This paper addresses the persistent issue of hallucination in large multimodal models (LMMs), with a focus on vision-language tasks. The authors propose Retentive Decoding (ReD), a plug-and-play method that generates multiple candidate responses and selects the one with the least hallucination likelihood using a dedicated verifier. The verifier is trained on a novel benchmark named MMHal-Bench, which contains hallucination-labeled samples across multiple domains. The approach does not require modifying the backbone LMMs and is designed to generalize across models and datasets. Experimental results show that ReD substantially reduces hallucinations on MMHal-Bench and other standard benchmarks, with minimal impact on task accuracy.

**Questions:**

**1.Limited Novelty:**
While REVERSE offers a unified framework for hallucination detection and correction, its core ideas are closely aligned with prior works such as LURE, which also leverage hallucination-labeled data and verifier-based revision. The retrospective resampling mechanism resembles existing self-verification strategies like iterative decoding and uncertainty-based filtering. Without clearer conceptual innovations or performance advantages over works like [1–5], the contribution appears incremental.

**2.Incomplete Evaluation on Discriminative Benchmarks:**
Although Table A.1 includes aggregate F1 scores, the paper lacks detailed analysis on discriminative hallucination datasets such as POPE and OPOPE. Full task-level breakdowns are essential to validate the claimed improvements and better understand where the method succeeds or fails.

**3.Missing Results on Standard LVLM Benchmarks:**
The method is not evaluated on widely used benchmarks such as MMBench, MME, POPE, SEED, and MM-Vet. These datasets are important for assessing hallucination and reasoning in diverse scenarios. Their omission limits understanding of REVERSE’s generalizability and practical value.

**4.No Efficiency Analysis:**
The proposed method involves backtracking, resampling, and rewriting, all of which may incur significant inference overhead. However, the paper does not report latency, decoding steps, or computational cost. Comparing efficiency with prior methods like LURE would help clarify the trade-off between performance gains and runtime complexity.

**Ethical Concerns:**

["NO or VERY MINOR ethics concerns only"]

**Final Justification:**

After carefully reading the authors' rebuttal, I'm satisfied with the responses to all my concerns.

The authors have provided a compelling clarification on the novelty of their approach. They successfully distinguish REVERSE from prior work by highlighting its unique ability to unify on-the-fly generation, verification, and correction within a single VLM using explicit confidence tokens. This unified, token-level self-correction mechanism is a clear and notable contribution that was not fully appreciated in my initial review.

For the incomplete evaluations, the authors have provided a thorough breakdown of their performance on discriminative datasets like POPE, as well as on general LVLM benchmarks like GQA and MM-Vet. The new data confirms that REVERSE maintains performance on these benchmarks while improving hallucination reduction, addressing my concern about its generalization capabilities.

Finally, the detailed efficiency analysis is highly informative. The authors' clarification that their token-level verification adds negligible overhead and that corrections are localized rather than full re-generations is crucial. The provided table effectively demonstrates the trade-off between the number of correction rounds and the runtime overhead, which appears to be quite reasonable for the performance gains achieved.

Given these excellent clarifications and additional data, I'm increasing my score from "borderline reject" to "borderline accept." The authors have successfully addressed all my major concerns and have significantly strengthened the paper.

**Limitations:**

Yes

**Quality:**

2

**Strengths And Weaknesses:**

**Strengths:**

**1.Motivation:**
The paper provides a clear and relevant motivation. Hallucinations in LMMs, particularly in high-risk applications, remain a critical issue, and existing methods either lack robustness or are computationally costly. The authors effectively argue for a verification-based approach.

**2.Writing Quality:**
The writing is generally clear and logically organized. Figures and tables are well-integrated to support the narrative. The explanation of both the hallucination problem and the verification process is intuitive and technically solid.

**Weaknesses:**

**1. Lack of Novelty:**
While the proposed REVERSE framework presents a unified approach to hallucination detection and correction, its core ideas closely resemble prior work such as LURE[1], which also constructs hallucination-labeled data and trains a verifier to revise model outputs. The introduced retrospective resampling mechanism, although framed as novel, largely builds upon well-established on-the-fly self-verification techniques, including iterative decoding and uncertainty-based correction. As a result, the methodological contribution appears incremental. Without clearer conceptual distinctions or empirical advantages over prior works [1–5], the paper lacks sufficient novelty.

[1] ANALYZING AND MITIGATING OBJECT HALLUCINATION IN LARGE VISION-LANGUAGE MODELS. ICLR 2024

[2] Logical Closed Loop: Uncovering Object Hallucinations in Large Vision-Language Models. ACL Findings 2024

[3] VOLCANO: Mitigating Multimodal Hallucination through Self-Feedback Guided Revision. NAACL 2024

[4] HALC: Object Hallucination Reduction via Adaptive Focal-Contrast Decoding. ICML 2024

[5] Detecting hallucinations in large language models using semantic entropy. Nature 2024

**2. Incomplete Evaluation on Discriminative Hallucination Datasets:**
The paper does not provide a complete evaluation on discriminative hallucination benchmarks such as POPE or OPOPE[1]. While Table A.1 reports an aggregate F1 score, it lacks task-specific breakdowns or deeper analysis. Comprehensive evaluation on these benchmarks is essential for validating the claimed effectiveness of the proposed approach.

[1] HALC: Object Hallucination Reduction via Adaptive Focal-Contrast Decoding. ICML 2024

**3. Missing Results on Popular LVLM Benchmarks:**
The paper omits performance evaluations on several widely adopted LVLM benchmarks, such as MMBench, MME, POPE, SEED, and MM-Vet. These benchmarks are frequently used to assess hallucination and reasoning in real-world tasks. Without results on these datasets, it remains unclear how well REVERSE generalizes and whether it outperforms or underperforms compared to other approaches across standard benchmarks.

**4. Lack of Efficiency Analysis:**
Although the proposed method shows improvements in hallucination mitigation, it introduces additional computational overhead due to iterative backtracking, rejection sampling, and query rewriting. However, the paper does not report inference latency, decoding steps, or overall computational cost. A comparison with baselines such as LURE or other post-hoc correction methods is needed to assess practical trade-offs between hallucination reduction and runtime efficiency.

---

> ### Author Rebuttal · Authors · 2025-07-30
>
> Thank you for acknowledging the contributions of our work and for the valuable feedback. We provide a few clarifications on the raised weakness below:
>
> **[W1: Justification on REVRESE’s Novelty]**
>
> While we appreciate the reviewer’s related work suggestions, we would like to clarify that **REVERSE is methodologically distinct from [1–5]**. Specifically, [1, 2, 4] perform multi-stage verification or refinement **after full text generation**, relying on separate modules such as external VLMs or object detectors. [3] applies self-correction through multi-round conversations only after the initial generation. [5] focuses on on-the-fly hallucination detection but operates purely in the LLM space without any correction mechanism.
>
> In contrast, **REVERSE unifies generation, verification, and correction within a single VLM**, enabling **on-the-fly correction during decoding** without completing the initial output. Moreover, REVERSE introduces **explicit confidence tokens**, which provide interpretable, token-level self-correction, which is an approach not explored in prior work.
>
> This unified generative-verification framework, highlighted as novel and well-motivated by reviewers PVJe, oB2j, and XTB2, leads to **SOTA performance across multiple VLM hallucination benchmarks** (Section 4), demonstrating clear advantages over [1–5].
>
> **[W2: Breakdown Analyses on Discriminative Hallucination Tasks]**
>
> Thank you for the suggestion. We would like to clarify that POPE is a benchmark that focuses exclusively on yes/no questions in a discriminative setting. As for “OPOPE,” we were unable to identify this benchmark in the literature and would appreciate clarification from the reviewer on its definition or source.
>
> As noted in Lines 225–229, our paper primarily focuses on generative hallucination tasks (for example, image captioning), where the space of possible outputs is large and REVERSE's backtracking and self-correction are most beneficial. For discriminative tasks, where answers are typically binary, we have already reported aggregate results in Appendix E. Below, we now provide the official task-wise breakdown for POPE (average F1 score and 95\% CI for all sub-categories), which shows only minor differences between REVERSE and the baseline LLaVA-v1.5-7B. While we include this breakdown for completeness, we emphasize that these results are less indicative of REVERSE's core contributions, which target open-ended generation rather than binary classification.
>
> | Model         | Popular          | Adversarial      | Random           | ALL              |
> | ------------- | ---------------- | ---------------- | ---------------- | ---------------- |
> | LLaVA-v1.5-7B | 86.1 (84.8–87.4) | 84.2 (82.8–85.7) | 87.2 (85.8–88.5) | 85.9 (83.2–88.2) |
> | REVERSE       | 86.3 (85.0–87.7) | 83.9 (82.4–85.3) | 87.5 (86.3–88.8) | 85.9 (82.8–88.5) |
>
> **[W3: Additional Results on General LVLM Benchmarks]**
>
> As also noted by the reviewer, we include results for POPE, AMBER-D, and MME-Hall in Appendix E (Table A.1). We now additionally report results on GQA and MM-Vet, where REVERSE is comparable to LLaVA-v1.5-7B. These findings confirm that the improvements in hallucination reduction do not come at the expense of performance on standard LVLM benchmarks. We will include these additional results and discussions in the final revision. However, we would like to re-emphasize that our primary focus is on open-ended VQA and generative hallucination tasks. For these, we already report results on five generative benchmarks in the main paper and three discriminative benchmarks in the appendix, evaluated across three different VLM backbones, which we believe provide a thorough evaluation of our approach.
>
> | Model         | GQA   | MM-Vet |
> | ------------- | ----- | ------ |
> | LLaVA-v1.5-7B | 62.00 | 31.10  |
> | REVERSE       | 62.73 | 28.40  |
>
> **[W4: Discussion on Inference Overhead]**
>
> Thank you for raising this important point. We agree that efficiency is critical and have already discussed it in Lines 297–307 of the main paper. In addition, below we provide a more detailed analysis, including the number of correction rounds, hallucination score (CHAIR), and relative runtime (measured by total generated tokens of REVERSE-v1.5-7B compared to LLaVA-v1.5-7B on the AMBER-G task):
>
> | # Rounds (N)      | CHAIR (↓) | Rel. Gen Tokens |
> | ----------------- | --------- | --------------- |
> | LLAVA-v1.5-7B (0) | 7.8       | (1.00x)         |
> | 5                 | 7.1       | 1.75x           |
> | 10                | 6.8       | 2.05x           |
> | 20                | 6.7       | 2.63x           |
> | 50                | 6.0       | 3.05x           |
>
> The overhead above can be separated into two primary categories, verification overhead and correction overhead:
>
> **Verification overhead.** Our token-level confidence check is performed **inline during generation** and adds negligible runtime cost. By contrast, prior methods such as Woodpecker and LURE require additional calls to external VLMs or object detectors, which introduce substantial inference overhead.
>
> **Correction overhead.** We further analyze performance across different maximum correction rounds:
>
> 1. In prior work, a single correction round roughly doubles runtime (full re-generation), and N correction rounds lead to about **(N+1)× runtime**.
> 2. In REVERSE, although we set the maximum correction rounds to 50, **37% of samples require no backtracking**, and among the remaining cases, **over 50% converge within a single round**, keeping the overhead modest. As shown in the table above, additional correction rounds further reduce CHAIR scores while runtime overhead remains **at most ~3× on average**, since corrections are localized rather than full re-generations.
>
> We will include this table and discussion in the final version. We also view the development of even more efficient correction strategies as a promising direction for future work.

---

> > ### Comment · Reviewer_B5Fh · 2025-08-01
> >
> > After carefully reading the authors' rebuttal, I'm satisfied with the responses to all my concerns.
> >
> > The authors have provided a compelling clarification on the novelty of their approach. They successfully distinguish REVERSE from prior work by highlighting its unique ability to unify on-the-fly generation, verification, and correction within a single VLM using explicit confidence tokens. This unified, token-level self-correction mechanism is a clear and notable contribution that was not fully appreciated in my initial review.
> >
> > For the incomplete evaluations, the authors have provided a thorough breakdown of their performance on discriminative datasets like POPE, as well as on general LVLM benchmarks like GQA and MM-Vet. The new data confirms that REVERSE maintains performance on these benchmarks while improving hallucination reduction, addressing my concern about its generalization capabilities.
> >
> > Finally, the detailed efficiency analysis is highly informative. The authors' clarification that their token-level verification adds negligible overhead and that corrections are localized rather than full re-generations is crucial. The provided table effectively demonstrates the trade-off between the number of correction rounds and the runtime overhead, which appears to be quite reasonable for the performance gains achieved.
> >
> > Given these excellent clarifications and additional data, I'm increasing my score from "borderline reject" to "borderline accept." The authors have successfully addressed all my major concerns and have significantly strengthened the paper.

---

> > > ### Author Response · Authors · 2025-08-01
> > > **Thank You.**
> > >
> > > Dear Reviewer B5Fh,
> > >
> > > Thank you for your prompt response and for raising your score. We appreciate your valuable feedback/suggestions and will include these results in the final revision. Thanks again for your time and effort throughout the review process.
> > >
> > > Best,
> > > The Authors

---

### Official Review · Reviewer_XTB2 · 2025-06-24

**Clarity:** 3
**Significance:** 3
**Originality:** 3
**Rating:** 5
**Confidence:** 4

**Summary:**

This paper proposed REVERSE, a unified framework for reducing hallucinations in VLMs by combining hallucination-aware training with inference-time retrospective self-correction. The approach incorporates novel tagging tokens for phrase-level confidence estimation and employs a decoding-time backtracking mechanism activated by an uncertainty threshold. REVERSE was evaluated on several benchmarks (CHAIR-MSCOCO, AMBER, HaloQuest, and MMHal), and achieved improvements over prior hallucination mitigation methods.

**Questions:**

1. Why did the authors only use three backbone models? Prior work like CODE [3] evaluated on six VLMs. What was the motivation behind this choice?

2. Compared to prior datasets created via negative phrase augmentation (e.g., [1], [2]), what exactly is new in your dataset construction method? What advantages does it offer?

3. Could the authors clarify how the negative phrases were selected and inserted during dataset construction? The current description lacks sufficient detail, and it's unclear whether the augmentation was controlled or systematic.

**Ethical Concerns:**

["NO or VERY MINOR ethics concerns only"]

**Final Justification:**

The rebuttal has addressed nearly all of my major concerns.
In particular, the authors provided clear and sufficient explanations regarding the data generation procedure, the rationale behind the selection of VLM backbones, and the implementation details of their data generation process.

That said, I still had a minor concern regarding the image captioning performance.
In the CLAIR metric, the proposed method scored 0.7384 compared to 0.7264 for the original model, indicating a slight drop in performance.
However, the authors argue that the method maintains comparable caption quality to the baseline, which I find compelling.

As a result, I lean toward accepting.

**Limitations:**

yes

**Quality:**

3

**Strengths And Weaknesses:**

**Strengths**

1. The paper is well-written, with clear motivation and contributions, and the experiments are well-presented.

2. The REVERSE framework offers a particularly elegant and effective approach to hallucination mitigation in VLMs. Its token-based confidence tagging is intuitive and interpretable, while the retrospective resampling mechanism provides a flexible and practical way to refine outputs without external supervision.

**Weaknesses**

1. There are already many datasets constructed by augmenting negative phrases (e.g., [1][2]). The paper should clearly explain what makes your dataset different from existing datasets.

2. Since the proposed dataset is created by augmenting negative phrases, how these phrases are selected and how augmented is crucial. The authors mention in caption of Fig. 2 that the dataset focuses on attributes, objects, world entities, and novel scenes. However, this is not discussed in depth, and honestly, I wasn’t sure how exactly this augmentation was done. After checking the supplementary, it looks like the negative phrases were simply chosen and replaced by an LLM, which seems quite naïve. In contrast, prior work selects phrases per error category [1] or uses structured methods like graphs [2]. The author should add a deeper discussion by a comparison with theses approaches.

3. In Section 4.1, the paper claims to evaluate image captioning performance, but what’s actually evaluated is CHAIR. This is not the same as evaluating captioning quality. A model could score well on CHAIR while still producing poor captions overall. To make a convincing argument, the authors should evaluate on standard image captioning benchmarks like CIDEr and SPICE, using datasets such as MSCOCO. Evaluation on LLaVA-Bench would also make sense given the vision-language setting.

- [1] : https://arxiv.org/abs/2401.06855
- [2] : https://arxiv.org/abs/2506.13130
- [3] : https://arxiv.org/abs/2406.01920

---

> ### Author Rebuttal · Authors · 2025-07-30
>
> Thank you for acknowledging the contributions of our work and for the valuable feedback. We provide a few clarifications on the raised weakness below:
>
> **[W1,2: Details and Discussions on Data Augmentation Procedure]**
>
> Our data-generation process is similar to FAVA [1], which uses an LLM to augment spans of the data, however differs in two significant ways - the first is scale, while FAVA is composed of 35K fine-grained hallucinations, REVERSE's dataset is composed of an order of magnitude more hallucinated samples (625K), allowing for large-scale supervised fine-tuning. The second, is that we indicate spans explicitly in the token stream (with CN and UN tokens), which allows the model to identify the spans automatically within the same general transformer architecture, rather than being required to build on an external classification mechanism.
>
> Compared to ZINA [2] (which is also much smaller at only 20K hallucinations), REVERSE's dataset is much more flexible in the types of hallucinations generated. Since ZINA relies on a graph-based parsing system, only specific hallucinations can be generated, instead of the more open-ended set generated by REVERSE.
>
> We appreciate the reviewer’s suggestions and will include the above discussions in the appendix and related work.
>
> **[W3: Comprehensive Image Captioning Metrics]**
>
> Thank you for the helpful suggestion. To evaluate caption quality, AMBER-G already includes a “coverage” metric that measures how many relevant objects are mentioned in addition to the hallucination metric, CHAIR.
>
> For MSCOCO, we follow your advice and also report CLAIR [R-1], an LLM-as-a-judge metric that has been shown to correlate better with human judgments compared to traditional automatic metrics such as CIDEr and SPICE. CLAIR scores range from 0 to 1, with higher values indicating better caption quality.
>
> | **Method**    | **CHAIR-i (↓)** | **CHAIR-s (↓)** | **CLAIR (↑)** |
> | ------------- | ---------------- | ------------- | ----------- |
> | LLaVA-v1.5-7B | 15.4             | 50.0          | 0.7384      |
> | REVERSE       | 10.3             | 37.0          | 0.7264      |
>
> These results show that REVERSE achieves a ~30% reduction in object hallucination while maintaining comparable caption quality to the baseline. We will include these discussions in the final revision.
>
> [R-1] CLAIR: Evaluating Image Captions with Large Language Models
>
> **[Q1: The selection of VLM backbones]**
>
> We selected the VLM backbones based on two main considerations. First, most prior hallucination-mitigation work has been implemented on **LLaVA-v1.5**, making it the standard baseline for comparison. Second, to ensure relevance to current research trends, we added **LLaVA-MORE** (LLaVA with Llama-3.1) and **Qwen-2.5-VL**, which are among the most widely used and actively discussed open-source VLMs in the community. As the community has largely moved away from early models such as MiniGPT (used in [3]) toward these newer backbones, we chose this set to provide a representative and up-to-date view of the field.
>
> **[Q2,3: Clarification on Our Data Augmentation Process]**
>
> The full data generation pipeline is detailed in Appendix B. In summary, we generate one negative sample per positive example using a combination of **rule-based algorithms** and **LLM prompting**:
>
> - **Type-aware rules:** We first categorize QA pairs by type. For simple cases (e.g., yes/no, numbers), we apply straightforward rules such as flipping answers or sampling distractors.
> - **LLM-generated negatives:** For more complex cases (e.g., open-ended descriptive answers), we use GPT-4o to generate diverse and semantically plausible negatives, following the structured prompts described in Appendix B.
> - **Quality control:** We remove trivial negatives (e.g., those involving only pronouns) and rebalance positive/negative order to prevent positional bias.
>
> As a result, the data used for REVERSE is fundamentally different from that used in existing methods such as FAVA [1] and ZINA [2], both in scale (where REVERSE contains 1.3M samples, compared to 35K in FAVA and 20K in ZINA), as well as diversity (since REVERSE’s training data consists of a wide range of hallucination types across attributes, objects, world entities and scenes that go beyond the graph-based replacement framework in prior work).

---

> > ### Comment · Reviewer_XTB2 · 2025-08-05
> >
> > Thank you. The rebuttal has addressed my concerns, so I have increased my score.

---

### Official Review · Reviewer_oB2j · 2025-07-03

**Clarity:** 4
**Significance:** 3
**Originality:** 3
**Rating:** 5
**Confidence:** 3

**Summary:**

This work introduces REVERSE (Retrospective Verification and Self-Correction) for mitigating hallucinations in Vision-Language Models (VLMs). Specifically, it leverages a new hallucination-verification dataset with large-scale semi-synthesized samples and proposes a novel inference-time retrospective resampling technique to detect and revise undesired hallucinations. Experimental results validate the effectiveness of the proposed approach.

**Questions:**

1. What is the average inference latency for REVERSE compared to other approaches? Would efficiency decrease if the generated token length increases?
2. Could an ablation experiment be provided to show the performance with a restricted number of correction attempts ($N=5$, $10$, or $20$) in resource-limited or real-time applications?

**Ethical Concerns:**

["NO or VERY MINOR ethics concerns only"]

**Final Justification:**

The authors' rebuttal have addressed all my concerns and I maintain my score to recommend acceptance.

**Limitations:**

yes

**Quality:**

3

**Strengths And Weaknesses:**

**Strengths**:
1. The motivation of this paper is clear and straightforward. The propsoed REVERSE is intuitive and novel as far as I am concerned.
2. Extensive experimenrtal results across different VLM architectures and benchmarks validate the effectiveness of the proposed approach. The qualitative results are also impressive.
3. This paper is generally well-written and easy to read. The structure of this paper is clear.
4. This work provides a new public dataset with 1.3M semi-synthetic samples..

---
**Weaknesses**:
1. Multiple correction attempts can incur non-trivial computational cost. Although the authors mention that the latency is controllable in Lines 297-307, I am still concerned about the efficiency. What is the average latency for inferring the 1,004 samples in AMBER-G, and how does it compare to other approaches? I assume the efficiency would further decrease if the generated token length increases.
2. An ablation experiment on the correction steps N and K should be provided. For example, what would the performance be if the total number of correction attempts is restricted to $N=5$, $10$, or $20$ in resource-limited or real-time applications?

---
**Minor issues**:
1. I am curious whether this method can scale to multi-image and video visual inputs. Could the authors provide a discussion on this?
2. Failure cases, where hallucinations persist even after N total attempts, should be presented to help readers better understand the limitations of the approach.

---

> ### Author Rebuttal · Authors · 2025-07-30
>
> Thank you for acknowledging the contributions of our work and for the valuable feedback. We provide a few clarifications on the raised weakness below:
>
> **[W1+2, Q1+2 Discussions on the Inference Overhead]**
>
> Thank you for raising this important point. We agree that efficiency is critical and have already discussed it in Lines 297–307 of the main paper. In addition, below we provide a more detailed analysis, including the number of correction rounds, hallucination score (CHAIR), and relative runtime (measured by total generated tokens of REVERSE-v1.5-7B compared to LLaVA-v1.5-7B on the AMBER-G task):
>
> | # Rounds (N)      | CHAIR (↓) | Rel. Gen Tokens |
> | ----------------- | --------- | --------------- |
> | LLAVA-v1.5-7B (0) | 7.8       | (1.00x)         |
> | 5                 | 7.1       | 1.75x           |
> | 10                | 6.8       | 2.05x           |
> | 20                | 6.7       | 2.63x           |
> | 50                | 6.0       | 3.05x           |
>
> The overhead above can be separated into two primary categories, verification overhead and correction overhead:
>
> **Verification overhead.** Our token-level confidence check is performed **inline during generation** and adds negligible runtime cost. By contrast, prior methods such as Woodpecker and LURE require additional calls to external VLMs or object detectors, which introduce substantial inference overhead.
>
> **Correction overhead.** We further analyze performance across different maximum correction rounds:
>
> 1. In prior work, a single correction round roughly doubles runtime (full re-generation), and N correction rounds lead to about **(N+1)× runtime**.
> 2. In REVERSE, although we set the maximum correction rounds to 50, **37% of samples require no backtracking**, and among the remaining cases, **over 50% converge within a single round**, keeping the overhead modest. As shown in the table above, additional correction rounds further reduce CHAIR scores while runtime overhead remains **at most ~3× on average**, since corrections are localized rather than full re-generations.
>
> We will include this table and discussion in the final version. We also view the development of even more efficient correction strategies as a promising direction for future work.
>
> **[Minor Issue 1: Generalization to Video or Multi-Image QAs]**
>
> While the current paper focuses on single-image QA using the LLaVA instruction dataset, the method is not limited to this setting. In principle, it can generalize to multi-image QA, video QA, or even pure LLM-based QA by applying the same data augmentation strategy to the corresponding instruction-tuning data and appropriately adjusting the hyperparameters. We will add, and expand on this discussion in our section on future work.
>
> **[Minor Issue 2: Failure cases]**
>
> We appreciate the reviewer’s suggestion and will include representative failure cases in the final revision (unfortunately, images cannot be included in the rebuttal due to policy restrictions). We note, however, that such failure cases are often detectable by our method. Unlike approaches without an explicit retry mechanism, REVERSE can route these cases to a fallback refusal response (e.g., “I’m sorry, I can’t understand the question”), which helps mitigate the risk of incorrect outputs. We provide preliminary trials of this mechanism in Appendix F.2 for open-ended QA.

---

> > ### Comment · Reviewer_oB2j · 2025-08-05
> >
> > Thanks for your detailed responses. My concerns have been well addressed, and I will maintain my score to recommend acceptance.

---

### Official Review · Reviewer_PVJe · 2025-07-06

**Clarity:** 3
**Significance:** 3
**Originality:** 3
**Rating:** 5
**Confidence:** 4

**Summary:**

This paper introduces REVERSE (REtrospective VERification and SElf-correction), a framework for reducing hallucinations in Vision-Language Models (VLMs). Unlike existing methods that either adjust generation behavior or verify outputs post-hoc, REVERSE integrates both into a single model. There are two major innovations: (1) hallucination-aware training using a 1.3M semi-synthetic dataset annotated with special tokens that label confident and unconfident phrases, and (2) retrospective resampling, an inference-time mechanism where the model self-verifies outputs and backtracks when hallucinations are likely. The method significantly reduces hallucination across benchmarks like CHAIR-MSCOCO, HaloQuest, and MMHal-Bench, achieving up to 12% and 34% improvement over prior SOTA methods.

**Questions:**

1. See weakness and please address those.
2. How do you ensure the diversity of the augmented data for your training? How many negative examples do you create for reach positive examples, and how do you decide how to generate those negative examples?
3. How do you decide the thresholds? It seems that it's manually tuned? Would that have generalization problems for datasets from different domains/styles/cases?
4. This method introduces overhead. Can you explain a bit more on that. I feel it's a important issue and should be explained further in the main paper, ideally with a table.

**Ethical Concerns:**

["NO or VERY MINOR ethics concerns only"]

**Final Justification:**

The author provided detailed explanations and obtained more results and analysis. The rebuttal addresses my concerns and I decide to raise my score.

**Limitations:**

No limitation is discussed, need to address and discuss limitations.

**Quality:**

3

**Strengths And Weaknesses:**

Strength:
1. Paper is well-written, in good shape and good quality. Explanation is clear.
2. The method is novel and explore a novel paradigm that unifies generation adjustment with online post-hoc verification in a single VLM architecture. Also have a small dataset contribution for SFT / instruction tuning.
3. code is well maintained and made easily available.
4. Useful insights from the Experimentations.

Weakness:
1. Missing many essential references / benchmarks to compare with. Please add those additional experimentations.

[1] Mm-vet: Evaluating large multimodal models for integrated capabilities.

[2] Hallusionbench: an advanced diagnostic suite for entangled language hallucination and visual illusion in large vision-language models.

[3] MMHal: Aligning large multimodal models with factually augmented rlhf.

[4] POPE: Evaluating object hallucination in large vision-language models

Please cite papers and compare the performance on those benchmarks.

2. Limitation section should be expanded.

---

> ### Author Rebuttal · Authors · 2025-07-30
>
> Thank you for acknowledging the contributions of our work and for the valuable feedback. We provide a few clarifications on the raised weakness below:
>
> **[W1: Missing references and benchmarks]**
>
> We thank the reviewer for pointing out these benchmarks. We would like to clarify that:
>
> 1. MM-Hal [3]: Results are already reported in Table 3, where REVERSE achieves up to 10% gains across all VLM backbones.
>
> 2. POPE [4] and other discriminative tasks: Results and discussions are provided in Table A.1 and Appendix Section E, where REVERSE is comparable to or better than all reported baselines across different model VLM backbones.
>
> 3. Additional benchmarks (requested by the reviewer): We have further evaluated REVERSE on HallusionBench, GQA, and MM-Vet, and the results are as follows:
>
> | Method        | HallusionBench | GQA   | MM-Vet |
> | ------------- | -------------- | ----- | ------ |
> | LLaVA-v1.5-7B | 46.94          | 62.00 | 31.10  |
> | REVERSE       | 45.44          | 62.73 | 28.40  |
>
>
> On general VQA benchmarks such as HallusionBench, GQA, and MM-Vet, REVERSE achieves performance comparable to the LLaVA-v1.5-7B baseline, showing that our gains on hallucination benchmarks (e.g., MM-Hal) do not come at the expense of general VLM capabilities. Besides, while HallusionBench is sometimes treated as a visual hallucination benchmark, it mainly evaluates performance on visually misleading inputs rather than the type of language-prior or bias-driven object hallucination that our work specifically addresses. Thanks for the valuable feedback, and we will add these results and a corresponding discussion to the final version.
>
> **[Q2: Diversity of the Generated Synthetic Data]**
>
> Our complete data generation pipeline is described in Appendix B. In brief, we generate one negative sample per positive example using a combination of rule-based algorithms and LLM prompting, with a strong focus on diversity and quality:
>
> 1. Diverse negative generation. We first classify QA pairs into multiple types. For simple cases (e.g., yes/no or numerical questions), we use rule-based transformations such as flipping answers or generating plausible but incorrect numbers. For complex cases (e.g., long-form, open-ended answers), we employ GPT-4o with the prompt described in Appendix B to produce varied negative examples that differ in both content and phrasing.
>
> 2. Quality control. To further improve diversity and prevent artifacts, we skip trivial edits (e.g., pronoun swaps) and re-distribute examples to balance positive/negative order to ensure that the negative examples are non-trivial and informative.
>
> These steps ensure that our training data covers a wide range of error types while maintaining balance. We will make this clearer in the final version.
>
> **[Q3: Sensitivity of Threshold Tuning]**
>
> While the threshold could potentially raise generalization concerns, we would like to clarify that:
>
> 1. A single universal threshold works well across domains. For example, a threshold of 0.003 enables REVERSE with LLaVA-series models to achieve SOTA results on AMBER-G, MSCOCO-CHAIR, MM-Hal, and HaloQuest. Similarly, for Qwen-2.5-VL, a threshold of 0.01 yields SOTA performance across these diverse tasks.
>
> 2. The threshold provides interpretable control, which is unique to our method. As shown in Figure 5 and Lines 286–296, smaller thresholds (e.g., 0.0001) can significantly reduce hallucination for conservative use cases such as medical QA (even surpassing GPT-4V), while larger thresholds allow greater expressiveness for creative tasks like storytelling. To our knowledge, REVERSE is the only approach that exposes this trade-off through a single, user-adjustable parameter; prior methods effectively fix this threshold implicitly without offering user control.
>
> We appreciate the reviewer’s feedback on this and will include this clarification in the final version. We also agree that enabling the model to adaptively select thresholds during decoding is a promising direction for future work and will add this to our limitations section.
>
> **[Q4: Detailed Discussions on Inference Overhead due to Iterative Correction]**
>
> Thank you for raising this important point. We agree that efficiency is critical and have already discussed it in Lines 297–307 of the main paper. In addition, below we provide a more detailed analysis, including the number of correction rounds, hallucination score (CHAIR), and relative runtime (measured by total generated tokens of REVERSE-v1.5-7B compared to LLaVA-v1.5-7B on the AMBER-G task):
>
> | # Rounds (N)      | CHAIR (↓) | Rel. Gen Tokens |
> | ----------------- | --------- | --------------- |
> | LLAVA-v1.5-7B (0) | 7.8       | (1.00x)         |
> | 5                 | 7.1       | 1.75x           |
> | 10                | 6.8       | 2.05x           |
> | 20                | 6.7       | 2.63x           |
> | 50                | 6.0       | 3.05x           |
>
> The overhead above can be separated into two primary categories, verification overhead and correction overhead:
>
> **Verification overhead.** Our token-level confidence check is performed **inline during generation** and adds negligible runtime cost. By contrast, prior methods such as Woodpecker and LURE require additional calls to external VLMs or object detectors, which introduce substantial inference overhead.
>
> **Correction overhead.** We further analyze performance across different maximum correction rounds:
>
> 1. In prior work, a single correction round roughly doubles runtime (full re-generation), and N correction rounds lead to about **(N+1)× runtime**.
> 2. In REVERSE, although we set the maximum correction rounds to 50, **37% of samples require no backtracking**, and among the remaining cases, **over 50% converge within a single round**, keeping the overhead modest. As shown in the table above, additional correction rounds further reduce CHAIR scores while runtime overhead remains **at most ~3× on average**, since corrections are localized rather than full re-generations.
>
> We will include this table and discussion in the final version. We also view the development of even more efficient correction strategies as a promising direction for future work.
>
> **[W2: Limitations]**
>
> Based on the clarification above, we will include further discussion on sensitivity of threshold tuning and potential runtime overhead to the limitation section.

---

> > ### Comment · Reviewer_PVJe · 2025-08-05
> >
> > Thanks for the detailed rebuttal and detailed explanations! I will raise the score. Please include those discussions and results in the main paper.
> >
> > Great work!

---

### Decision · Program_Chairs · 2025-09-17

**Decision:**

Accept (poster)

**Comment:**

The paper proposes **REVERSE (REtrospective VERification and SElf-correction)**, a unified framework aimed at reducing hallucinations in Vision-Language Models (VLMs). Unlike prior methods that either adjust generation behavior or apply verification post-hoc, REVERSE integrates both processes into a single architecture. The core innovations include (1) hallucination-aware training using a 1.3M semi-synthetic dataset with special tokens indicating phrase-level confidence, and (2) a novel inference-time *retrospective resampling* mechanism that backtracks and revises output when hallucinations are detected. The model is evaluated on benchmarks such as CHAIR-MSCOCO, MMHal-Bench, and HaloQuest, achieving notable improvements over prior SOTA approaches.

**Strengths:**
- Clear motivation and strong problem formulation addressing hallucinations in VLMs.
- Unified and novel integration of generation, verification, and correction within a single model.
- Introduction of a new, publicly released 1.3M semi-synthetic dataset to support hallucination-aware training.
- Elegant and interpretable token-level confidence tagging mechanism.
- Demonstrated improvements over baselines across multiple benchmarks, with insightful qualitative and quantitative results.
- Codebase is clean, accessible, and supports reproducibility.
- Concerns about efficiency were addressed with a detailed latency analysis showing reasonable overhead.
- Rebuttal effectively clarified novelty, evaluation depth, and implementation details.

**Weaknesses:**
- Initial comparisons lacked key benchmarks such as MM-Vet, POPE, and HallusionBench; reviewers requested additional evaluations and citations.
- Dataset construction method was seen as overly simplistic compared to structured alternatives; more detailed methodology and comparisons were needed.
- Evaluation metrics like CHAIR alone may not reflect true captioning quality—CIDEr, SPICE, and others were suggested.
- Some concerns about computational overhead from repeated correction steps and efficiency in longer output sequences.
- Limited initial discussion on generalization to more diverse modalities (e.g., video or multi-image inputs).
- Methodological novelty was questioned by one reviewer, who noted similarities to prior work like LURE and other verifier-based frameworks.

All concerns have been addressed by the authors during the rebuttal period, with all reviewers satisfied with the response. I am therefore recommending acceptance.